# Selective $CO_2$ reduction to $CH_3OH$ over atomic dual-metal sites embedded in a metal-organic framework with high-energy radiation

Changjiang Hu[1,8], Zhiwen Jiang [1,8], Qunyan Wu[2,8], Shuiyan Cao[3], Qiuhao Li[1], Chong Chen[1], Liyong Yuan[2], Yunlong Wang [1], Wenyun Yang[4], Jinbo Yang [4], Jing Peng[5], Weiqun Shi [2], Maolin Zhai[5] ✉, Mehran Mostafavi [6] ✉ & Jun Ma [1,7] ✉

The efficient use of renewable X/γ-rays or accelerated electrons for chemical transformation of $CO_2$ and water to fuels holds promise for a carbon-neutral economy; however, such processes are challenging to implement and require the assistance of catalysts capable of sensitizing secondary electron scattering and providing active metal sites to bind intermediates. Here we show atomic Cu-Ni dual-metal sites embedded in a metal-organic framework enable efficient and selective $CH_3OH$ production (~98%) over multiple irradiated cycles. The usage of practical electron-beam irradiation (200 keV; 40 kGy min⁻¹) with a cost-effective hydroxyl radical scavenger promotes $CH_3OH$ production rate to 0.27 mmol g⁻¹ min⁻¹. Moreover, time-resolved experiments with calculations reveal the direct generation of $CO_2^{•-}$ radical anions via aqueous electrons attachment occurred on nanosecond timescale, and cascade hydrogenation steps. Our study highlights a radiolytic route to produce $CH_3OH$ with $CO_2$ feedstock and introduces a desirable atomic structure to improve performance.

The combustion of fossil fuels has provided convenient energy for centuries, but the finite extent of accessible supplies and the climate change linked to atmospheric carbon dioxide ($CO_2$) accumulation present a solid motivation to seek alternative energy sources. Technologies that exploit diverse renewable power to convert $CO_2$ into fuels are being developed to supplement the imperative energy demands and provide solutions for carbon neutrality. Among $CO_2$ reduction products, methanol ($CH_3OH$), a liquid with a low risk of flammability, promises to be one of the leading substitutes for fossil fuels and could be distributed with existing petroleum infrastructure and used with fuel cells to generate electricity[1,2]; however, the $CO_2$-to-$CH_3OH$ route, which involves the transfer of six electrons and six protons, typically suffers from high kinetic barriers and low selectivity[3–5]. One major source for the high activation energy corresponds to the change in orbital hybridization and geometry that $CO_2$ experiences upon the first electron attachment, which presents a

[1]Department of Materials Science and Technology, Nanjing University of Aeronautics and Astronautics, Nanjing 211106, P. R. China. [2]Laboratory of Nuclear Energy Chemistry, Institute of High Energy Physics, Chinese Academy of Sciences, Beijing 100049, P. R. China. [3]College of Physics, Nanjing University of Aeronautics and Astronautics, Nanjing 211106, P. R. China. [4]State Key Laboratory for Mesoscopic Physics, School of Physics, Peking University, Beijing 100871, P. R. China. [5]Radiochemistry and Radiation Chemistry Key Laboratory of Fundamental Science, College of Chemistry and Molecular Engineering, Peking University, Beijing 100871, P. R. China. [6]Institut de Chimie Physique, UMR8000 CNRS/Université Paris-Saclay, 91405 Orsay, France. [7]School of Nuclear Science and Technology, University of Science and Technology of China, Hefei, Anhui 230026, P. R. China. [8]These authors contributed equally: Changjiang Hu, Zhiwen Jiang, Qunyan Wu. ✉e-mail: mlzhai@pku.edu.cn; mehran.mostafavi@universite-paris-saclay.fr; junma@nuaa.edu.cn

perennial challenge to conventional thermal-, light-, and electrochemical-driven $CO_2$ reduction. In contrast, high-energy radiation (X/γ-rays or accelerated $e^-$) ionizes and excites water, resulting in an initial homogeneous distribution of abundant reactive radicals[6]. The short-lived hydrated electron ($e_{aq}^-$), which represents the most effective reducing species known in nature, activates $CO_2$ to form $CO_2^{\bullet-}$ radicals with an almost diffusion-controlled rate (Eq. 1)[7].

$$e_{aq}^- + CO_2 \rightarrow CO_2^{\bullet-}; k_1 = 8.2 \times 10^9 M^{-1} s^{-1} \qquad (1)$$

This first activation process has been regarded as the most energy-demanding and rate-limiting step in conventional $CO_2$ reduction using photolytic and electrolytic electrons, but it can be readily achieved via radiolytic aqueous electrons[8]. By virtue of $e_{aq}^-$ chemistry, we have previously shown a selective and catalyst-free $CO_2$ conversion to oxalate through rapid $CO_2^{\bullet-}$ dimerization at ambient aqueous conditions (Eq. 2)[9,10].

$$CO_2^{\bullet-} + CO_2^{\bullet-} \rightarrow C_2O_4^{2-}; k_2 = 1.4 \times 10^9 M^{-1} s^{-1} \qquad (2)$$

Thus, the transformation of $CO_2^{\bullet-}$ radicals to $CH_3OH$ has prospected to be further extended by coupling controllable intermediates with precisely designed catalysts.

From a practical perspective, ionizing radiations are secure, cost-effective, and compatible with renewable and nuclear energy sources. For decades, high-energy systems, including accelerated electrons and $^{60}Co$ γ-ray, have been used as the essence of modern wastewater or flue gas treatment[11,12], medical[13], and material processing[14,15]. Commercial $e^-$ accelerators have witnessed a sharp global rise in numbers and can be powered by excessive electricity from renewable solar, hydro, and wind energy sources; $^{60}Co$ γ-ray sources made by neutron bombardment have constituted a mature product from nuclear reactors. Under safety regulations, the irradiation equipment and facilities can be coupled with the output of industrial waste $CO_2$ sources or remotely used to treat captured $CO_2$. Besides, $CO_2$ is the major carbonaceous component of other planetary atmospheres, so an all-day operation of $CH_3OH$ production using emitted rays from daily-discharged radioactive waste may develop exciting scenarios for long-lasting spaceship fuel cells and prebiotic chemistry, which are rarely accessible by existing techniques.

Therefore, as illustrated in Fig. 1, this process is anticipated that sustainable radiation energy will be converted into fuels with $CO_2$, a renewable C1 feedstock, and then stored as chemical bonds; however,

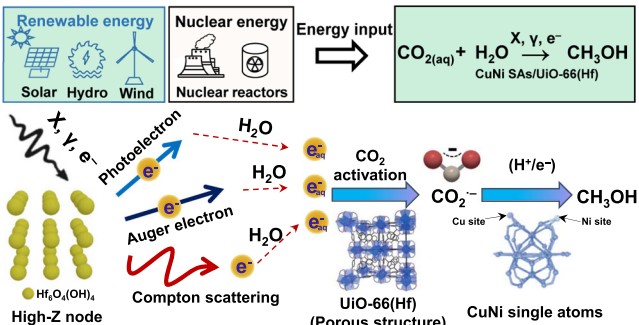

**Fig. 1 | Schema of radiation-catalyzed $CO_2$-to-$CH_3OH$ conversion.** High-energy radiation that compatible with renewable energy sources (solar, hydro, and wind energy) and additional nuclear energy sources (nuclear reactors) is used to drive the process, in which CuNi SAs/UiO-66(Hf) acts as radiosensitizer and intermediates modulator. Figure 1. Crystal structure of UiO-66(Hf) (porous structure) and CuNi single atoms. These structural figures were generated using the Diamond software (Version 4.6.1) developed by Crystal Impact GbR. Reference: Diamond– Crystal and Molecular Structure Visualization, Crystal Impact–Putz, H., Brandenburg, K., **102**, 53227 Bonn, Germany, https://www.crystalimpact.de/diamond.

the development of effective catalysts for generating $CO_2^{\bullet-}$ radicals and converting them into $CH_3OH$ as much as possible is crucial to the success of this strategy. In irradiated water, one drawback is that the number of electrons per energy absorbed is limited to ~$2.8 \times 10^{-7}$ mol $J^{-1}$ (G-values) due to the low scattering cross-section and the oxidation with parent •OH radicals. In this regard, we rationalize that the newly discovered radiosensitizers, nano-sized metal organic-framework (MOFs), are deemed the best candidates[16–19]. These zeolite-like materials are constructed by stitching together inorganic poly-nuclear clusters [termed secondary building units (SBUs)] and organic linkers by strong bonds, featuring exceptionally high specific area and porosity. SBUs can be easily designed with Hf- and Zr-based high-Z metal nodes/clusters to display a high probability of interacting with incoming ionizing radiation[20]. In solutions, the unique nanoscale 3D arrays of SBUs greatly enhance the scattering of Auger/Compton electrons during a cascade of ionizing events and lead to additional production of low-energy electrons (LEEs; 0−20 eV) (Fig. 1)[21]. LEEs could interact with SBUs subsequently because each SBU is surrounded by other high-Z metal clusters extending in all directions, resulting in a chain reaction to generate additional electrons. Besides, the cumulative porous architectures of MOFs enable them to have excellent mass transfer characteristics and increase the local reactants to expedite the initial activation. Thus, the topologies of MOFs, the ideal integration of 3D arrays of ultrasmall metal-oxo clusters with high porosity, make them the leading materials to maximize $CO_2^{\bullet-}$ generation under irradiation.

$CO_2^{\bullet-}$ anion radicals, once formed in solutions, are likely to undergo disproportionation to formate and CO or recombine to oxalate[9,10], so another major challenge is to direct them toward $CH_3OH$, which can suppress the yield of undesired by-products. Copper (Cu) is the best-known metal for binding $CO_2^{\bullet-}$ radicals to form *OCHO or *COOH intermediates (* denotes the surface-coordinated state), providing a catalytic origin to produce $CH_3OH$[22]. Cu-based single-atoms (SAs) embedded in MOF matrices have recently demonstrated exceptional activity and product selectivity because of the homogeneous coordination structures and unique electronic properties of metal centers[23–25]. MOFs can be a platform for single-atom site separation, stabilization, and functional realization[22–27]. The defect sites of the MOF scaffold, namely UiO-66, can anchor atomic Cu sites and prevent individual metal atoms or tiny clusters from migrating and sintering during the catalytic reaction, which accounts for the superior performance. When a transition metal promoter like nickel (Ni) is introduced, dual-metal-sites are formed in their respective single-site forms. This structure can cooperate to optimize interactions between the adjacent active sites and reactants on the catalyst surface, minimizing the overall endothermic energy of essential C1 intermediates (*CO、*COOH、*CHOH、*CH$_2$OH) in the $CO_2$ reduction process[28–30]. Therefore, we took the view that developing Cu-based atomic dual-metal-sites implanted MOFs could increase and tune the $CO_2^{\bullet-}$ conversion—would play a decisive role in the practical feasibility of radiolytic $CO_2$ reduction to $CH_3OH$, as well as in the atomistic understanding of the process.

In this work, we develop a catalytic strategy to convert $CO_2$ into $CH_3OH$ using a high-energy radiation technique combined with atomic engineering of MOFs-based catalysts, which differs from the existing thermochemical, electrolytic, and photolytic techniques. As a proof-of-concept, the $^{60}Co$ γ-ray irradiation of $CO_2$-saturated aqueous solutions with atomic CuNi SAs/UiO-66(Hf) leads to effective $CH_3OH$ production under ambient conditions. When a •OH scavenger was used, we achieved $CH_3OH$ selectivity of ~98% and energy conversion efficiency of ~$1.5 \times 10^{-7}$ mol $J^{-1}$, which breaks the limitation of radical yield in neat water radiolysis. UiO-66(Hf)-based catalysts display resistance to γ-ray and maintain the activity across multiple irradiation cycles. Specifically, the remarkable catalytic production rate (~0.27 mmol $g^{-1}$ min$^{-1}$) was obtained by high-dose-rate electron

beam irradiation. Pulse radiolysis revealed transient species and nanosecond kinetics of $CO_2^{\cdot-}$ radicals, which played a critical role in identifying the binding motifs of atomic Cu-Ni dual-metal-sites during the radiolytic reaction. Besides, diffuse reflectance infrared Fourier transform spectroscopy (DRIFTS) experiments, and DFT calculations suggested that the active sites stabilize the various C1 intermediates and minimize undesired by-products, leading to high $CH_3OH$ selectivity. The new controllable $CO_2$ reduction integrating single-atom catalysts with a radiolysis methodology could provide an effective solution for $CO_2$ emissions reduction and sustainable energy storage.

## Results and discussion

### Catalyst design and fabrication

The selection of MOFs and synthesis strategies has various priorities according to specific catalytic processes. For instance, single-atoms (SAs) for photocatalysis are primarily prepared using light-responsive MOFs with the conjugated π-system[22,26,31,32]. Conductive carbon materials derived from the pyrolysis of MOFs are often preferred for electrocatalysis[23,33,34]. In this study, the radiation-driven catalytic process requires that MOFs sensitize the ionizing radiations and stabilize SAs while avoiding degradation[35,36]. In investigating possible support materials, we discovered UiO-66(Hf), one of the most easily accessible MOFs, exhibited qualified radiation resistance and robust binding capacities for SAs, as well as presenting record-setting high probabilities of cascading secondary electron scattering[16,21,37]. To incorporate the dual-metal-sites into UiO-66(Hf), we again use the irradiation reduction approach to match the catalytic environments. $e_{aq}^-$ and $\cdot$H radicals can also easily reduce metal ions to the zero-valent state and give rise to metal clusters in a possible matrix. The formation of metal atoms and progressive coalescence of atoms for clusters was first observed in the 1990s[38]. Since then, it has emerged as an effective and scalable strategy to achieve numerous mono- and multi-metallic clusters and nanocomposite materials. Figure 2a shows that the dual-metal-sites implanted UiO-66(Hf) was easily prepared via initial hydrothermal fixation of metal ions precursors ($Cu^{2+}/Ni^{2+}$) at 85 °C, followed by an ambient radiolytic reduction (More details are given in experiment section). Inductively coupled plasma atomic emission spectroscopy (ICP-AES) measurements revealed that CuNi SAs/UiO-66(Hf) contained 1.6 and 0.33 wt% Cu and Ni, respectively (Supplementary Table 1), and a Brunauer-Emmett-Teller surface area of ~750 m² g⁻¹ (Supplementary Fig. 1a and Supplementary Table 2). The crystalline UiO-66(Hf) typically has a porous framework with a porosity radius of ~2.1 nm. The dispersion of CuNi SAs on UiO-66(Hf) matrixes slightly increases the porosity volume and $CO_2$ and $N_2$ uptake

(Supplementary Fig. 1a and 1b). This observation is attributed to the coordinated water or impurity decomposition via γ-rays irradiation, which has been reported previously[39]. The powder X-ray diffraction (PXRD) patterns indicated the well-retained crystalline structure before and after γ-ray irradiation up to 4 kGy (Supplementary Fig. 2). Moreover, transmission electron microscopy (TEM) demonstrated the presence of Cu and Ni single sites on the UiO-66(Hf), as well as the absence of large aggregates of metallic particles (Fig. 2b)[22]. In addition, the elemental mapping from energy dispersive spectroscopy (EDS) suggested that Cu, Ni, and Hf were uniformly distributed throughout the skeleton of the support. In UiO-66(Hf), one SBU is coordinated by 6 Hf atoms via 12 benzene-1,4-dicarboxylate (BDC) linkers. Unlike other MOFs, missing linker defects and missing cluster/$Hf_6$ node defects of UiO-66(Hf) commonly exist, and the integrity of the structure can be well maintained after the linker or even cluster is missing. NMR and Neutron diffraction results (Supplementary Fig. 3a and Fig. 3b) revealed that the ligand-contributed atoms occupied ~92% of UiO-66(Hf), and indicated the presence of ligand defects.

To disclose the local coordination structure, element-selective X-ray absorption fine structure (XAFS) measurements were conducted. Figures 3a, b, Supplementary Fig. 4 to Supplementary Fig. 7, and Supplementary Table 3 illustrated the Cu K-edge and Ni K-edge X-ray absorption near-edge structure (XANES) spectra of CuNi SAs/UiO-66(Hf) with the reference spectra of CuO, Cu foil, NiO, and Ni foil. In SAs catalysts, the valence state of metal often behaves as $\delta^+$ ($M^{\delta+}$) due to its unsaturated coordination structure. The absorption Cu K-edge position of CuNi SAs/UiO-66(Hf) was located between that of Cu foil and CuO, suggesting that the valence state of Cu atoms was positively charged and between 0 and 2, which was in accordance with the results in X-ray photoelectron spectroscopy spectra (XPS) (Supplementary Fig. 8). The Cu 2$p$ XPS peaks in CuNi SAs/UiO-66(Hf) at 932.8 eV for Cu 2$_{p3/2}$ and 952.5 eV for Cu 2$_{p1/2}$, which can be assigned to Cu⁺ [22,40]. In contrast to Cu, the Ni 2$p$ XPS peak of CuNi SAs/UiO-66(Hf) at 856.1 eV in Supplementary Fig. 9 appeared to be similar to the typical Ni²⁺ (856.5 eV)[32,41,42], which coincided with Ni K-edge. The absorption edge of Ni was closer to that of NiO. These spectral data suggested that the valance of Ni prefers Ni²⁺, and $e_{aq}^-$ was more accessible to reduce Cu²⁺ than Ni²⁺.

The extended X-ray absorption fine structure spectrum (EXAFS) was obtained by taking the Fourier transform of XAFS data to acquire further insights into the coordination structure of metal atoms Cu²⁺ and Ni²⁺. As shown in Fig. 3e, the prominent peak of Cu was at 1.44 Å in CuNi SAs/UiO-66(Hf). Compared with the spectra of CuO and Cu foil, the distinctive absorption peak revealed that the Cu element exists as the single-atom form, coordinating with oxygen sites provided by

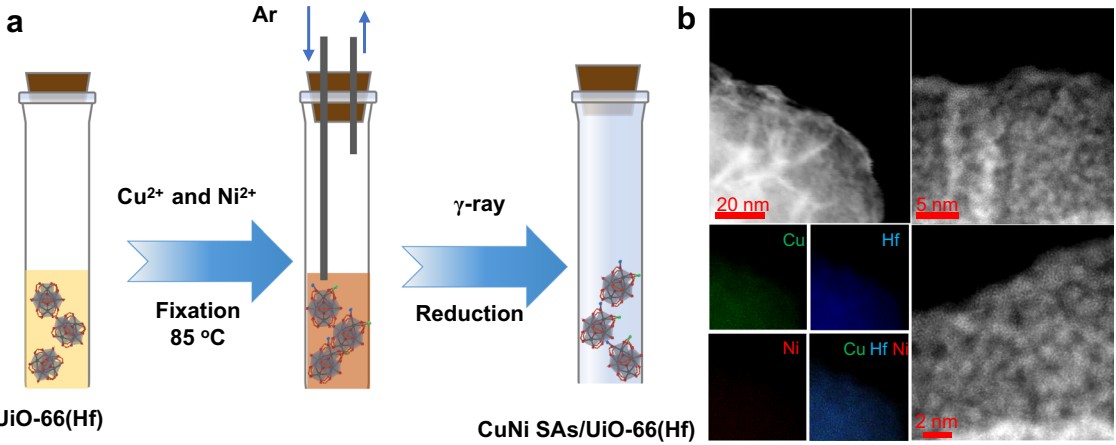

**Fig. 2 | Synthesis and characterization of UiO-66(Hf) MOF-supported CuNi dual-atom catalysts through radiation-driven reduction strategy. a** The synthesis scheme of CuNi SAs UiO-66(Hf). **b** TEM and EDS mapping of CuNi SAs/UiO-66(Hf). Hf (blue), Cu (green), Ni (red).

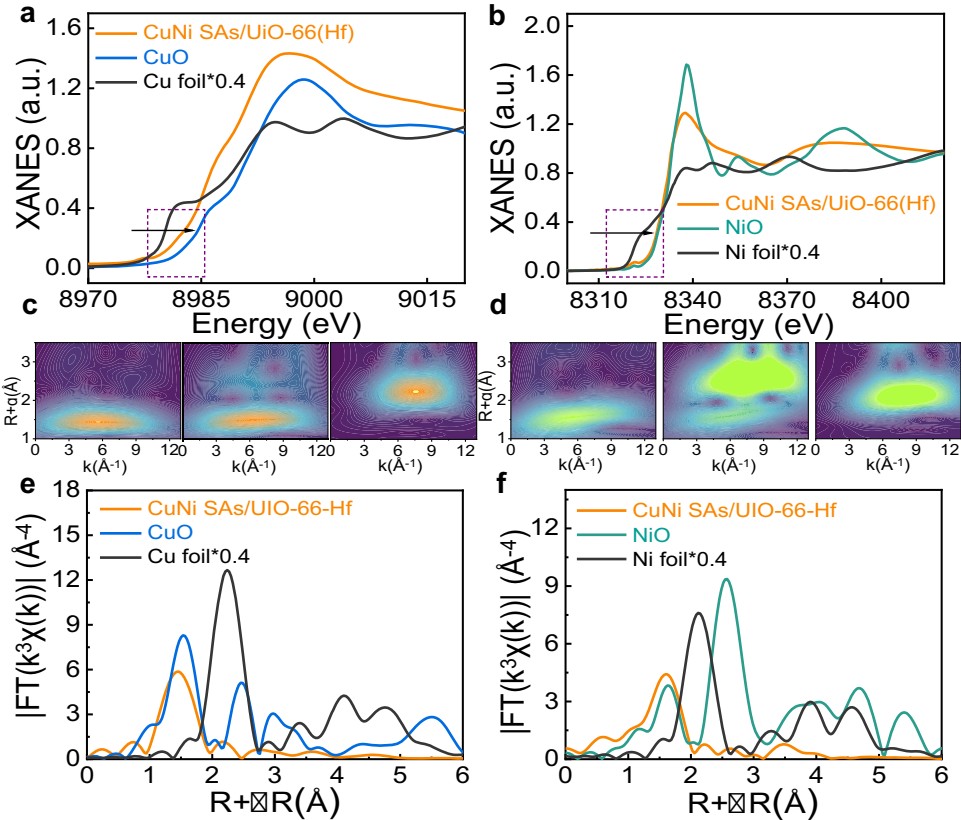

**Fig. 3 | The local coordination structure of CuNi SAs/UiO-66(Hf). a** Normalized XANES results of the samples at the Cu K-edge. **b** Normalized XANES results of the samples at the Ni K-edge. **c** WT-EXAFS of the CuNi SAs/UiO-66(Hf) samples, CuO and Cu foil. **d** WT-EXAFS of the CuNi SAs/UiO-66(Hf) samples NiO, and Ni foil. **e** EXAFS results of the samples at the Cu K-edge. **f** EXAFS results of the samples at the Ni K-edge. The a.u. stands for arbitrary units. Source data are provided as a Source data file.

UiO-66(Hf), which is also consistent with the wavelet transform (WT) data. WT studies of Cu EXAFS confirmed that one intensity maximum at 5.3 Å$^{-1}$ in CuNi SAs/UiO-66(Hf) was attributed to the Cu-O bond (Fig. 3c). Similarly, only one intensity maximum at 5.6 Å$^{-1}$ and a peak of Ni at 1.60 Å were acquired (Fig. 3d, f). As a result, only Cu-O or Ni-O bonds were found by XANES data analysis in CuNi SAs/UiO-66(Hf), and there is no other bonding form for these two metals. Thus, these observations confirmed no direct interactions or bonding between the Ni and Cu atoms in the bimetallic single-atom samples.

## Catalytic activities driven by $^{60}$Co γ-ray/electron beam

The catalytic CO$_2$ reduction activity was first evaluated in CO$_2$-saturated aqueous solutions under $^{60}$Co γ-ray irradiation (Fig. 4a). The gaseous and liquid products evolution increases linearly with absorbed dose, so the radiolytic yield expressed as *G*-values is measured to qualify the efficiency. Blank experiments (a reactor with argon-saturated solutions) were carried out to ensure no catalysis decomposition under γ-ray irradiation and found negligible carbonaceous products. As a control in catalyst-free CO$_2$-saturated solution, the emission of CO product is detected due to CO$_2$$^{\bullet-}$ radical disproportionation as previous reports[7,9]. The γ-ray initial reactions with a UiO-66(Hf) generate a higher CO yield. As described above, UiO-66(Hf) can effectively scatter the secondary electrons during an ionizing event[16,21]. We rationalized that such electron sensitization in porous structures would accelerate initial CO$_2$ activation to CO$_2$$^{\bullet-}$. The slight increase of CO release with *G*-values of $0.08 \times 10^{-7}$ mol J$^{-1}$ was also observed in respective Cu or Ni SAs/UiO-66(Hf) cases, implying the SAs metal sites play a specific role in stabilizing CO$_2$$^{\bullet-}$. Notably, taking two SAs metals together, the bimetallic monatomic catalysts generate CH$_3$OH with a *G*- value of $0.1 \times 10^{-7}$ mol J$^{-1}$ and promote CO yield up to

$0.22 \times 10^{-7}$ mol J$^{-1}$ (Supplementary Fig. 10). Although the product selectivity is unsatisfactory ~32.4%, these preliminary data suggested that the radiolytic CO$_2$-to-CH$_3$OH routine is feasible with the assistance of CuNi SAs/UiO-66(Hf).

Water radiolysis generates reductive e$_{aq}$$^-$, $\cdot$H radicals, and the highly oxidizing hydroxyl radical, $\cdot$OH. The accumulated $\cdot$OH with prolonged irradiation likely hindered the subsequent reactions of reduced intermediates such as hydrogenation[9]. The γ-ray catalytic activities were further performed to avoid this effect in 0.01 M Na$_2$SO$_3$ solutions containing the •OH scavenger base on the following Eq. 3[43].

$$SO_3^{2-} + {}^{\bullet}OH \rightarrow SO_3^{\bullet-} + OH^- \tag{3}$$

Among many potential $\cdot$OH scavengers, our work found the best efficacy with Na$_2$SO$_3$ (Fig. 4a). Sulfite is a cost-effective chemical and could be readily produced through the flu gas SO$_2$ removal process on an industry-scale[44]. The practicality of the radiolytic approach thus can potentially be reinforced by integrated treatment for CO$_2$ and SO$_2$, the major component of industrial exhaust gas. In this case, we found that a tiny amount of CuNi SAs/UiO-66(Hf) (0.05% weight percentage) displayed exceptional conversion efficiency and CH$_3$OH selectivity (97.8%). The trace of other products, such as CH$_4$, C$_2$H$_4$, and formic acid, is below the detection limit of the chromatography (Supplementary Fig. 11 to Supplementary Fig. 13). The *G*-value of CH$_3$OH production was promoted from 0.1 to $0.9 \times 10^{-7}$ mol J$^{-1}$. Because the CO$_2$-to-CH$_3$OH conversion occurs with the transfer of six electrons and six protons, the obtained *G*(CH$_3$OH) value far exceeds the e$_{aq}$$^-$ values ($2.8 \times 10^{-7}$ mol J$^{-1}$) in pure water and those with isolated Cu and Ni SAs/UiO-66(Hf). Furthermore, Fig. 4b suggested that the CH$_3$OH formation

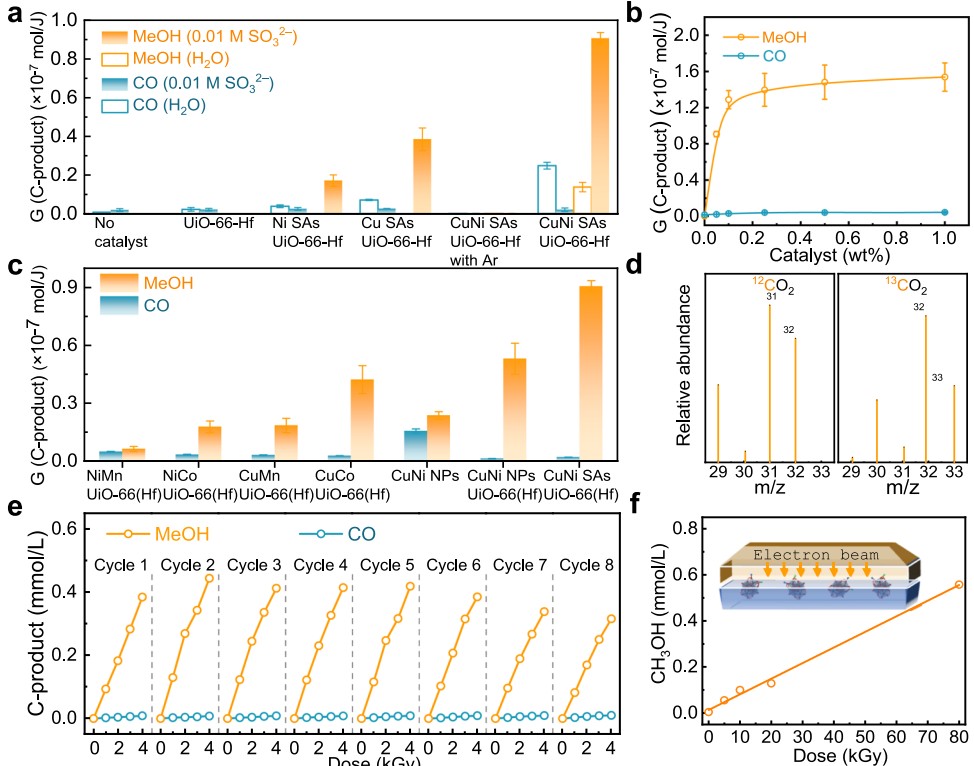

**Fig. 4 | $^{60}$Co γ-ray/electron-beam driven catalytic CO$_2$-to-CH$_3$OH performance in various conditions. a** $G$-value of MeOH and CO of 0.05 wt% with catalyst UiO-66(Hf), Ni SAs/UiO-66(Hf), Cu SAs/UiO-66(Hf), and CuNi SAs/UiO-66(Hf) in water or 0.01 M Na$_2$SO$_3$ solution with CO$_2$ saturated under 1–4 kGy γ-ray irradiation. **b** The amplified of C-product in $G$-value by the increase of CuNi SAs/UiO-66(Hf) mass fraction. **c** Catalytic activities comparison of various dual-metal-sites under identical condition. **d** Mass spectra of $^{13}$CH$_3$OH ($m/z$ = 33) and $^{12}$CH$_3$OH ($m/z$ = 32) produced in the radiation catalysis reduction of $^{13}$CO$_2$ or $^{12}$CO$_2$ over CuNi SAs/UiO-66(Hf). **e** Cycle test of 0.05 wt% CuNi SAs/UiO-66(Hf) in water with CO$_2$ saturated solution under 1–4 kGy γ-ray irradiation. **f** Catalytic CH$_3$OH production via electron beam (200 keV; 40 kGy/min) irradiation. Error bars represent the standard deviation from at least three independent measurements. Source data are provided as a Source data file.

was raised to $1.5 \times 10^{-7}$ mol J$^{-1}$ by increasing the catalyst mass fraction from 0.05 to 0.2 wt%.

We found that atomic CuNi catalysts are superior to NPs. Especially for metal NPs like Cu, their size-dependent electronic structures were reflected in their radiation-induced catalytic activity. UiO-66(Hf) supported CuNi NPs composites (Supplementary Fig. 14) were synthesized based on the reported solvent-thermal methods[32]. Their performance in CH$_3$OH production was conducted at identical irradiated conditions, and the results are shown in Fig. 4c. With dispersed CuNi NPs alone, we found that the radiolytic approach still can produce CH$_3$OH; however, both the total yield and selectivity were far lower than the values with CuNi NPs/UiO-66(Hf) and CuNi SAs/UiO-66(Hf). On the other hand, CuNi NPs/UiO-66(Hf) displays a comparable selectivity, while the efficacy is only about 60% that of CuNi SAs/UiO-66(Hf). Furthermore, we have synthesized various dual-metal-sites embedded in UiO-66(Hf), i.e., Cu/Co, Cu/Mn, Ni/Co, Ni/Mn. The catalytic activities in Fig. 4c showed that other bimetallic materials could not grant comparable capability during high-energy radiation catalysis. However, the high selectivity observed in series of Cu/Ni, Cu/Co, and Cu/Mn suggested the important role of metallic copper. The comprehensive data imply that metallic Cu may extend the lifetime of CO$_2$·$^-$ radicals through the initial surface stabilization process.

In all conducted experiments, no carbonaceous compounds were found in Na$_2$SO$_3$ solutions saturated with argon gas; the $^{13}$CO$_2$ labeling experiments with mass spectrometer analysis confirmed the origins of products (Fig. 4d). We demonstrated that the catalysts attenuated <15% in the CH$_3$OH production activity over 8 cycles of irradiation experiments, suggesting their qualified stability and viability in practical use (Fig. 4e). After the stability test, we conducted TEM, EDS

mapping, and XPS on the used samples to investigate whether any structural transformations occurred. The characterizations (Supplementary Fig. 8, Supplementary Fig. 9, and Supplementary Fig. 15) showed that the skeleton structure of UiO-66(Hf) support remains preserved, confirming the durable radiation-resistance property. However, XPS data found that the protic solvent water can gradually leach Ni atoms from UiO-66(Hf) over several cycles. The ICP results indicate that the Ni element content is reduced to 0.015% after 8 cycles. This observation may be one of the key factors contributing to performance degradation.

CO$_2$-to-CH$_3$OH conversion involves proton-coupled 6e$^-$ transfer. The cumulative electrons from six-folds $G$(CH$_3$OH) = $0.9 \times 10^{-7}$ mol J$^{-1}$ achieved by CuNi SAs/UiO-66(Hf) were nearly double $G$(e$_{aq}^-$) ~ $2.8 \times 10^{-7}$ mol J$^{-1}$ in water radiolysis. Substantial evidence showed that high atomic-number (Z) element clusters enhanced interaction with the cascading secondary electrons in 3D SBU arrays, leading to efficient radiolytic water splitting and radiation therapy[16–18,20,21]. In UiO-66, SBUs composed of Hf-metals could absorb much more energy possessed by the secondary electrons and produce electrons, which collide with the adjacent SBU again, resulting in significant chain reactions. UiO-66(Hf) afforded superior radiosensitization over Zr-UiO-66 and conventional nanoparticles such as HfO$_2$ by more efficiently scattering secondary electrons, leading to the formation of additional low-energy electrons (LEEs; 0–20 eV). Subsequently, the increased interaction between LEEs and confined H$_2$O provided more precursors of solvated electrons, and thus rapid electron transfer in UiO-66(Hf) is responsible for effective CO$_2$ reduction to give CH$_3$OH[17,19]. The assumption was validated by electron quenching studies using NO$_3^-$ and Cd$^{2+}$ ions, in which the products

were eliminated (Supplementary Fig. 16). On the other hand, $CO_2$ can be adsorbed by MOF that is often associated with: (i) two parallel aromatic rings with interatomic spacings of ~7 Å; (ii) metal-oxygen-metal bridges; and (iii), open metal sites[21]. UiO-66(Hf) also enhances the adsorption of aqueous $CO_2$ through the coordination of metal atoms with water clusters in the pores[45]. This Hf-O-C-O adsorption model facilitates the transfer of electrons from the metal to $CO_2$. It is likely that a similar model has been adopted for single-atomic sites adsorbed with metal (Cu/Ni) M-$CO_2^-$, thus providing an electron transfer channel from UiO-66(Hf) to $CO_2$ molecules.

Our earlier attempt showed that the use of a small quantity (<0.2 wt%) of UiO-66(Hf)-OH can achieve a γ-rays-to-hydrogen conversion efficiency exceeding 10% that significantly outperforms Zr-/Hf-oxide nanoparticles and the existing radiolytic $H_2$ promoters[21]. This work was conducted with 0.05 wt% CuNi SAs/UiO-66(Hf) in $CO_2$-saturated aqueous solution under 4−32 kGy γ-ray irradiation. It is estimated, for instance, with an absorbed dose of 10 kGy, 0.5 g catalysts in 1 liter water would produce 0.9 mmol $CH_3OH$. The production rate is another important issue. As electron beams output a large amount of energy in a short time with increasing electrical-to-electron conversion efficiency, the high-dose-rate electron beam was selected to evaluate the $CH_3OH$ production rate. We achieved a remarkable rate of 0.55 mmol $L^{-1}$ $CH_3OH$ with only 2 min of irradiation for 80 kGy, a competitive value to the existing photolytic and electrolytic methods (Fig. 4f).

Although irradiation pathways facilitate conversion at room temperatures due to the formation of reactive intermediates and progress has been made during the 1970s−1990s[46−49], it should also be emphasized that the conventional radiolytic $CO_2$ conversion pathways suffered from the side reactions, resulting in low radical yields and poor selectivity for specific products, in which CO is unavoidable in certain fraction. Now, we have developed fundamentally new

chemistry that offers a creative solution to the challenge of recycling $CO_2$ into energy-rich $CH_3OH$ with water. This work found that combining high-energy radiation with contemporary catalysts, i.e., CuNi SAs/UiO-66(Hf), can overcome the long-standing challenges of selectivity and energy efficiency. To the best of our knowledge, there are no reports so far on the selective and effective production of $CH_3OH$ or any liquid fuels based on the high-energy system.

## Transient kinetics of $CO_2^-$ radicals at nanosecond timescale

Currently, we are suggesting an emerging radiolytic catalysis concept, which is rooted in basic radiation chemistry, aiming to solve the most pressing environmental and energy challenges. While chemical transformations via radiation and catalysis are individually well-developed and optimized in many cases, efficient and effective radiolytic catalysis coupling remains primitive. In this context, a time-resolved understanding of $CO_2$ activation, intermediates binding, hydrogenation, and eventual $CH_3OH$ production at the irradiated interface is indispensable. Radiolytic $CO_2$ reduction involves generating electrons and $CO_2^{\bullet-}$ as the intermediates (Supplementary Fig. 17 and Fig. 5a); therefore, efficient $CH_3OH$ production requires a full understanding of the activation step and $CO_2^{\bullet-}$ binding motifs on a single-atom basis. Although some theoretical prediction and experimental evidence of surfaces bounded $CO_2^{\bullet-}$ were given[50,51], most spectroscopic methods are inadequate to resolve the kinetics occurring on the timescale from nanoseconds to microseconds. To alleviate this shortcoming, we use pulse radiolysis to measure the $CO_2^{\bullet-}$ dynamics as a function of active metal sites and UiO-66(Hf) support. This approach enables probing critical steps in their reactions, allowing the assembly of detailed mechanisms of how the catalysts function.

Figure 5 presented the transient absorption profiles in solution in the absence and presence of various UiO-66(Hf)-based samples at 600 ns. To focus on $CO_2^{\bullet-}$ transformation, the mechanistic study uses

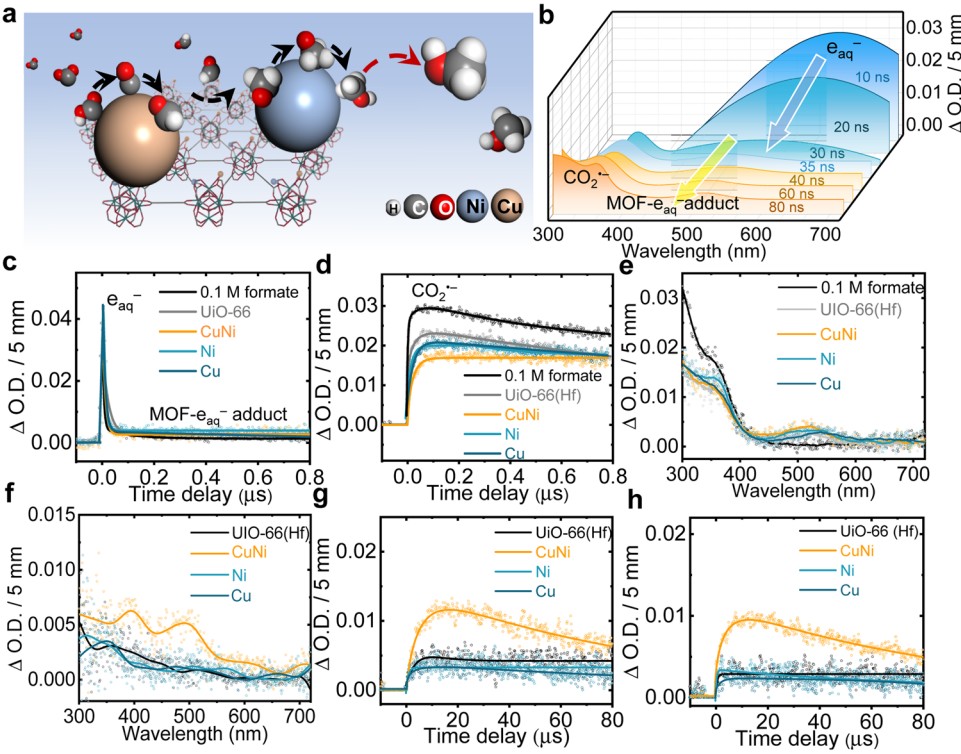

**Fig. 5 | Pulse radiolysis measurement in $CO_2$-saturated 0.1 M formate solution in the absence and presence of various catalysts. a** Schematic diagram of $CO_2$ reduction with 0.25 mg/mL CuNi SAs/UiO-66(Hf) catalyst. **b** 3D stereographs of transient absorption spectra at different time in UiO-66(Hf) dispersion.

**c, d** Transient absorption kinetic traces at 520 nm (**c**) and 360 nm[57] (**d**). **e** Transient absorption spectrum at 600 ns. **f** Transient absorption spectrum at 60 μs. **g, h** Transient absorption kinetic traces at 390 nm (**g**) and 500 nm (**h**). Source data are provided as a Source data file.

formate to scavenger $\cdot$OH and $\cdot$H into $CO_2^{\bullet-}$ radical (Eqs. 4 and 5).

$$\cdot OH + HCOO^- \rightarrow CO_2^{\bullet-} + H_2O \qquad (4)$$

$$\cdot H + HCOO^- \rightarrow CO_2^{\bullet-} + H_2 \qquad (5)$$

The transient absorption spectra in $CO_2$-saturated solution containing UiO-66(Hf) were shown in Fig. 5b. After 10 ns of the electron beam pulse, a characteristic broad peak appeared in the range of 400–720 nm, which is ascribed to $e_{aq}^-$ specifically in the bulk phase. In the vicinity of 40 ns, the peak of $e_{aq}^-$ gradually diminished, and a new absorption peak around 520 nm was observed, indicating the formation of intermediate species correlated with $e_{aq}^-$ decay. The reference spectrum was measured in Ar-saturated 0.1 M tert-butanol solutions, showing no $CO_2^{\bullet-}$ is formed and only $e_{aq}^-$ remained because of the scavenging reaction of $\cdot$OH and $\cdot$H radical with tert-butanol (Supplementary Fig. 18). Comparing with a reference spectrum, the distinct peak at 520 nm is likely assigned to adducts complex formed via the reaction between UiO-66(Hf) and $e_{aq}^-$. Although the sites for electron attachment are unknown, we speculate that the rapid electron delocalization process occurred within the MOF skeleton. In addition, the transient kinetics (Fig. 5c) in different UiO-66(Hf) samples further confirmed the formation of these adducts accompanied by the drastic decay of $e_{aq}^-$.

At the timescale from 40 to 80 ns, the typical transient absorption spectrum of $CO_2^{\bullet-}$ radical was also readily identified in the 300–400 nm region. The transient kinetics at 360 nm found $CO_2^{\bullet-}$ increased rapidly within 100 ns and stabilized for hundreds of nanoseconds (Fig. 5d). Among the systems with different MOF samples, the decay of $CO_2^{\bullet-}$ at 360 nm in the absence of UiO-66(Hf) displayed slight variation with UiO-66(Hf) alone but exhibited noticeable slower decay with UiO-66 samples embedded with Cu or Ni metal sites. The mitigated decay showed that the recombination reaction of $CO_2^{\bullet-}$ radicals was suppressed and gave birth to the intermediate state of absorbed

$CO_2^{\bullet-}$ radicals predominantly on metal sites rather than UiO-66(Hf). Of particular importance is that dual-metal CuNi SAs/UiO-66(Hf) exerted better stabilization of $CO_2^{\bullet-}$ than respective single-site forms. The decay difference revealed the synergistic interaction between dual-metal-sites towards $CO_2^{\bullet-}$. As the time reached 600 ns (Fig. 5e), the characteristic peak for $CO_2^{\bullet-}$ radicals and the electron-adduct complex remained stable. The above nanosecond transient processes conclude that the dual-metal-sites are responsible for the adsorption of $CO_2^{\bullet-}$ radicals, while the MOF skeleton reacts with $e_{aq}^-$ and anchors them in the SBU readily for further reduction.

Surprisingly, transient signals on a prolonged time scale (60 μs) suggested that dual-metal-sites resulted in a peculiar reduction pathway for absorbed $CO_2^{\bullet-}$ radicals with respect to UiO-66(Hf) and single-site forms (Fig. 5f–h). A pair of intense peaks at 390 nm and 500 nm in the dual-metal spectrum appeared in contrast to other absorption spectra (Fig. 5f). The concerted decay of both peaks characterized the same intermediate species attributed to the further reduction of $CO_2^{\bullet-}$ radicals, which did not occur in UiO-66(Hf) and its single-site forms (Fig. 5g, h). This critical feature of the difference revealed that the potential selective reduction of $CO_2^{\bullet-}$ radicals only took place in dual-metal-sites, which is consistent with the irradiation catalytic performance.

### Hydrogenated intermediates

The evolution of hydrogenated intermediates like *COOH and CH3O* is critical to the overall CH3OH production. We conducted DRIFTS spectral measurements under deep UV irradiation (193 ~ 248 nm). The results in Fig. 6a clearly showed the formation of surface-bound *COOH and CH3O* within minutes. The transient peaks of these intermediates are consistent with previous reports[28,52]. Fortunately, $e_{aq}^-$ could be formed through the excitation by UV light in $Na_2SO_3$ solution[43], and the band gap of UiO-66 was also in line with the energy of deep UV light to produce the photoelectron for $6H^+/6e^-$ reduction of $CO_2$-to-CH3OH. As shown in Fig. 6a, the peak of *$CO_2$ and *$HCO_3$ as carbon dioxide adsorption species was located at 1692 cm$^{-1}$ and

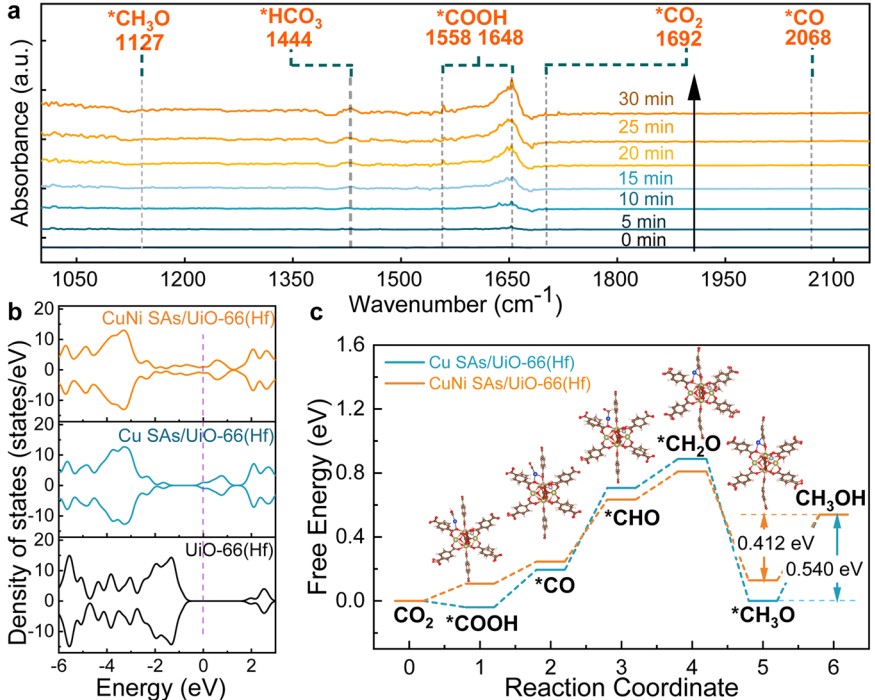

**Fig. 6 | DRIFTS and DFT calculations of $CO_2$ reduction. a** In situ DRIFTS spectra for detecting the reaction intermediates during $CO_2$ reduction over CuNi SAs/UiO-66(Hf) under deep UV irradiation. **b** The density of states for Cu-Ni SAs/UiO-66(Hf), Cu SAs/UiO-66(Hf), and UiO-66(Hf). **c** Free energy profiles of $CO_2$ to CH3OH. The a.u. stands for arbitrary units. Source data are provided as a Source data file.

1444 cm$^{-1}$, which may act as active bodies on the catalyst surface to provide a source of CH$_3$OH. The *COOH absorption band at 1558 cm$^{-1}$ and 1648 cm$^{-1}$ gradually increased along with the increase in illumination time, revealing the hydrogenation of CO$_2^-$ radicals or CO$_2$ on single atom sites. Furthermore, the formation of CH$_3$O* was suggested by the stretching vibration band at 1127 cm$^{-1}$, which represents the crucial intermediates directly related to the CO$_2$ to CH$_3$OH. It should be noted that an in-situ DRIFTS setup under γ-ray irradiation has not been developed so far. The highly penetrating γ-ray irradiation often causes the malfunction of electronic devices, making it almost impossible to position the instrument close to the $^{60}$Co γ-ray sources. Instead, we are currently updating a custom-designed DRIFTS setup coupled with X-rays. We anticipate these ongoing efforts will provide further insights into radiolytic CO$_2$ reduction and relevant radiation catalysis processes.

In order to better illustrate the hydration pathway CO$_2$ reduction to CH$_3$OH over Cu-Ni dual-metal active sites and the electronic structure of original CuNi SAs/UiO-66(Hf), DFT theoretical calculations were carried out with VASP packages. Supplementary Fig. 19 depicted the four different coordination modes of Cu and Ni bimetallic elements. Based on the relative energy of the four optimized structures, Type I is the most stable one, of which two oxygen atoms of the carboxylic acid with adjacent Hf coordinate with Cu and Ni atoms, respectively, and the Ni and Cu atom shares a common plane. The fitting data of XANFES provided by the structure of Type I is in good agreement with measured spectra, in which the Ni and Cu atom is separated by 5.6 Å with no interactions.

To verify the electron transfer effect of bimetallic CuNi SAs/UiO-66(Hf) catalyst, Fig. 6b the calculated total density of states (TDOS) of the bimetallic CuNi SAs/UiO-66(Hf), single Cu SAs/UiO-66(Hf) and UiO-66(Hf) are presented in Fig. 6b, compared with the single Cu SAs, a narrower band gap was shown in the TDOS of bimetallic Cu-Ni SAs, which clarifies the faster electron excitation and transferred to intermediate groups from SBU. Supplementary Fig. 20 depicts the role of Ni monatomic addition in the two-component single-atom catalyst. The results demonstrated that Ni atoms effectively occupied the lower molecular orbital, reducing the bandgap. Based on the results of the TDOS, the bimetallic monatomic catalyst of CuNi SAs/UiO-66(Hf) accelerates electron migration, enabling the six-electron reduction of CO$_2$ to CH$_3$OH.

The calculated reaction pathway of 6H$^+$/6e$^-$ reduction of CO$_2$ to CH$_3$OH unfolds in Fig. 6c. The intermediate states agree with DRIFTS data. It is known that the *COOH was provided in homogeneous reactions, and the final product was influenced by pH conditions to obtain formic acid and oxalic acid with polycarboxylic acid. The addition of CuNi SAs effectively induced the adsorption of *COOH at the site, which was then converted to *CO. Consistently, the addition of SAs catalyst did not result in forming formic acid or oxalic acid, confirming that CO$_2^{\cdot-}$ was adsorbed to form *COOH on the metal centers. The *COOH intermediate should be quickly formed *CO but not generate CO. *CO easily undergoes hydrogenate to methoxy. These are the two crucial factors for CH$_3$OH formation[53]. Compared with Cu SAs, CuNi SAs catalyst showed little effect on the free energy of *COOH to *CO and other steps (*CO to CHO and *CHO to *CH$_2$O). However, as shown in Fig. 6c, the last two steps of proton transfer were the decisive steps for a better performance of CO$_2$-to-CH$_3$OH with a diatomic SAs catalyst. We speculated that three intermediate adsorption states might occur at the Cu site and the last two at the Ni site, consistent with the CO$_2$-to-CH$_4$ pathway[29]. The partial density of states (PDOS) doping to Ni sites concluded that Ni sites enabled *CH$_2$O to transfer electrons faster and accelerate the proton hydrogenation of *CH$_2$O to form *CH$_3$O. It is analogous that the free energy was also reduced to 0.412 eV via Eq. 6.

$$*CH_3O + H^+ + e^- \rightarrow CH_3OH + * \tag{6}$$

Interestingly, both final steps were enhanced at the bimetallic site, reflecting that the CuNi SAs/UiO-66(Hf) enables a more efficient and selective conversion of CO$_2$ to CH$_3$OH.

In conclusion, according to the performance and the long-term stability, a promising strategy has been proposed for using renewable high-energy radiation to convert CO$_2$ into value-added chemicals, thereby storing renewable energy (solar, wind) and nuclear energy into chemical energy. We showed that the atomic design of dual-metal sites embedded in UiO-66(Hf) plays a critical role in the strategy's effectiveness. UiO-66(Hf) matrix contributed as a platform for SAs stabilization and promoting ionizing radiation conversion efficiency, while atomic dual-metal-sites account for the high product selectivity due to their unique capability to capture CO$_2^{\cdot-}$ anions. We applied pulse radiolysis to uncover the underlying mechanism by observing the nanosecond kinetics of aqueous electrons and CO$_2^{\cdot-}$ anions radicals in irradiated catalytic solutions. The spectral data and DFT calculations suggested that the three C1 intermediate adsorption states might occur at the Cu site and the final two at the Ni site, primarily responsible for selective CH$_3$OH formation. The cycling test and electron beam experiment further strengthen the practicality of the radiolytic approach. The present study provides a practical solution to tackle CO$_2$ emissions and energy storage, as well as offers insights into the interfacial CO$_2$ reduction processes on a dynamic and atomic basis.

## Methods

### Chemicals

Dimethylformamide (DMF, 99%, Macklin), methanol (99.5%, Macklin), ethanol (99.7%, Macklin), formic acid (99%, Aladdin), oxalic acid (99%, Aladdin), hafnium (IV) chloride (HfCl$_4$, 99.5%, Macklin), terephthalic acid (BDC, 99%, Macklin), sodium sulfite (98%, Aladdin). All chemicals were used without further purification. Ultrapure water with a resistivity of 18.25 MΩ cm was obtained from a water purification system. CO$_2$ (99.999%) and Ar (99.999%) were purchased from Air Liquide Industrial Gases Company or Nanjing Shangyuan Industrial gas plant.

### Characterization

X-ray diffraction (XRD) analysis was performed using Rigaku D/MAX-RB with a scan speed of 4°/min. ICP spectroscopy was recorded by an Agilent 7500Ce spectrometer. Energy dispersive spectroscopy (EDS) was employed to determine elemental composition. Transmission electron microscopy (TEM) and high-resolution transmission electron microscopy (HRTEM) images were recorded by a JEOL JEM-2010F microscope combined with mapping. X-ray photoelectron spectroscopy (XPS) measurements were carried out on a VG Multi Lab 52000 system with a monochromatic Al Ka X-ray source (Thermo VG Scientific). The XAFS spectra data were collected at the BL1W1B station in the Beijing Synchrotron Radiation Facility (BSRF, operated at 2.5 GeV with a maximum current of 250 mA), and the Athena and Artemis software package provides the EXAFS fitting results.

### Synthesis of UiO-66(Hf)

The solvothermal reaction was performed by dissolving HfCl$_4$ (0.3 g) and BDC (terephthalic acid, 0.174 g) into a mixture of 10 mL DMF and 0.018 mL H$_2$O. The final mixture was transferred into a Parr Teflon lined stainless steel vessel (23 mL) and heated at 100 °C for 24 h. After the reaction mixture was slowly cooled to room temperature, the obtained powder was collected and washed by DMF and methanol and vacuum-dried overnight at 80 °C.

### Synthesis of various dual-metal sites on UiO-66(Hf)

The obtained UiO-66(Hf) sample was first immersed in methanol for 3 days, and methanol was replaced once every 12 h, and then vacuum dried for 24 h at 160 °C. Typically, 60 mg UiO-66(Hf) and 10 mg CuCl$_2$, and 120 mg NiCl$_2$·6H$_2$O in 50 acetonitrile (MeCN) were heated at 85 °C with stirring for 2 h. The resulting product (yellow powder) was

washed with MeCN, then dried in a vacuum overnight at 80 °C, followed by irradiation reduction for 8 h in 0.1 M isopropanol ((CH$_3$)$_2$CHOH) aqueous solution. External high-energy $^{60}$Co γ-ray radiolysis experiments were carried out using a $^{60}$Co source with a dose rate of 0.36 kGy h$^{-1}$ at the Nanjing University of Aeronautics and Astronautics Radiation Center. The absorbed dose was calibrated using a Fricke dosimeter. The obtained powder was collected and washed by MeCN and vacuum-dried overnight at 80 °C.

The single atoms Ni SAs/UiO-66(Hf) and Cu SAs/UiO-66(Hf) were obtained through a similar procedure by changing CuCl$_2$ to NiCl$_2$·6H$_2$O. The single atoms CuNi SAs/UiO-66(Hf) was obtained through a similar procedure using CuCl$_2$ and NiCl$_2$·6H$_2$O.

The CuMn, CuCo, NiMn, NiCo/UiO-66(Hf) was prepared following a similar synthetic procedure to that of CuNi SAs/UiO-66(Hf), except for using 120 mg MnCl$_2$ or CoCl$_2$·6H$_2$O and 10 mg CuCl$_2$ 120 mg MnCl$_2$ or CoCl$_2$·6H$_2$O and 120 mg NiCl$_2$·6H$_2$O.

CuNi NPs/UiO-66(Hf) was synthesized via the NaBH$_4$-based thermal reduction of the above-mentioned yellow powder. The powder (20 mg) was first dispersed in methanol. Then, 1 mL methanol solution containing 2 mg NaBH$_4$ was introduced into the suspension solution and vigorously stirred to produce CuNi NPs/UiO-66(Hf). After stirring for 30 s, the resulting black solid was washed with methanol and dried under a vacuum at room temperature.

### G(C-product) calculation

$$G_{tot}(CH_3OH) = \frac{n(CH_3OH)}{w_{aq}\gamma_{aq} + w_M\gamma_M} \quad (7)$$

$$G_{aq}(CH_3OH) = \frac{n(CH_3OH)}{w_{aq}\gamma_{aq}} \quad (8)$$

Where n(CH$_3$OH) is the amount of CH$_3$OH produced by radiolysis, $w_{aq}$, and $w_M$ are the mass fraction of water and catalyst in the mixture, and $\gamma_{aq}$ and $\gamma_M$ are the respective absorbed doses of the water and catalyst. However, in our experiment, the weight of the catalyst ranged from 0.05 to 0.1%, much less than that of H$_2$O content, so the contribution to the total absorbed dose was ignored. The value of $\gamma_{aq}$ was measured by the Fricke dosimeter and finally used as the total absorbed dose.

### Selectivity calculation

The selectivity to the specific product is defined as the amount of product produced in terms of carbon divided by the amount of all carbon-containing products in terms of carbon.

$$Selectivity = \frac{Moles\ of\ carbon\ product}{Moles\ of\ all\ carbon - containing\ products} \quad (9)$$

### $^{60}$Co γ-ray/electron beam-driven catalytic CO$_2$ reduction experiments

The γ-ray catalytic CO$_2$RR experiment was finished in a 40 mL penicillin bottle using a panoramic $^{60}$Co source with a dose rate of 0.36 kGy h$^{-1}$ at room temperature. In a typical experimental process, 10 mg of CuNi SAs/UiO-66(Hf) was dispersed into 20 mL deionized water or 0.01 M Na$_2$SO$_3$ solution. Then, the mixture is saturated with high-purity carbon dioxide or argon. Gas and liquid products were detected by gas chromatography with TCD and FID and liquid chromatography, respectively. The liquid chromatography did not detect products such as formic acid and acetic acid.

During the experiment on methanol synthesis using an electron beam, 50 mg catalyst was dispersed in a 50 mL solution of Na$_2$SO$_3$ (0.01 mol L$^{-1}$) saturated with CO$_2$. The mixture was then transferred to a 200 mm petri dish and irradiated using a 200 keV electron beam (MEB-200 Zhiyan Technology Co., LTD). This experimental setup

proved more effective than an open system and prevented the collection of gas products. Notably, the production rate of carbon monoxide (CO) was negligible under γ-ray irradiation, thus leading to the exclusion of CO production from further analysis in this study.

### Pulse radiolysis experiments

Pulse radiolysis experiments employed the picosecond laser-triggered electron accelerator, ELYSE, coupled with a time-resolved absorption spectrophotometric detection system. Laser (260 nm) driven Cs$_2$Te photocathode allows the production of short electron pulses with a typical half width of 7 ps, a charge of ~6 nC, and an energy of ~7.8 MeV at a repetition rate of 10 Hz. During irradiation, the sample solutions were contained in a fused cell with a path length of 5 mm, connected to a closed circulation system from a 100 mL stock solution that was used to renew the sample in the irradiation cell after each pulse using a peristaltic pump (flow rate: 100 mL/min). The diameter of the electron beam was 3 mm, and the irradiated volume was <0.1 mL.

Absorption spectral measurements were performed using the white light from a homemade Xenon flash lamp. The light was focused on the sample parallel to the electron beam with a smaller diameter and then directed onto a flat field spectrograph (Chromex 250IS), which disperses the light on the entrance optics of a high dynamic range streak-camera (model C-7700-01 from Hamamatsu) to obtain an image resolved in wavelength and time. The kinetic data and absorption spectra were extracted from three series of 200 resulting images. In this work, the transient spectra were measured from 290 to 720 nm at 1 μs and 100 μs.

### Dissolution $^1$H NMR Spectroscopy

The sample was prepared by weighing 10 mg of the UiO-66(Hf) into an NMR tube. Subsequently, 1 mL of resolution solution (1 M NaOH in D$_2$O) was added to the tube, followed by ultrasound for 30 s. The $^1$H NMR Spectroscopy was collected by Bruker Avance III 400 MHz NMR.

### Neutron powder diffraction

The data were collected at the China Institute of Atomic Energy (CIAE) using the Peking University High-intensity Powder Diffractometer (PKU-HIPD). To enhance the quality of the data, we introduced deuterated benzene-1,4-dicarboxylic acid as the ligand during the synthesis process while keeping the other steps unchanged. This modification allowed us to obtain improved data for further analysis and characterization. In order to analyze the NPD patterns, a combined Rietveld refinement method was utilized. The refinement process used a step size of 0.05° for the NPD and was carried out using the FULLPROF suite program.

### DRIFT experiments

10 mg samples were placed in an in-situ infrared reaction (Harrick diffuse accessory), and 0.08 mL of water was added. Initially, residual air in the reactor was purged using argon gas, followed by injecting 99.999% CO$_2$ into the sample. The Xe lamp (CEL-HXUV300-T3) source provided the light source with a cutoff filter to achieve UV light irradiation (<380 nm), and the infrared spectrum was collected under continuous illumination for 30 minutes with a step length of 500 ms.

### Computational methods

The density functional theory (DFT) calculations were carried out using the projector augmented wave (PAW)[54] method as implemented in the Vienna Ab Initio Simulation Package (VASP)[55]. The electronic exchange-correlation potential was described by the Perdew-Burke-Ernzerhof generalized gradient approximation (PBE-GGA)[56]. Spin-polarization calculations were performed with the plane-wave cutoff energy of 500 eV and using the Gaussian smearing with a smearing value of 0.2 eV. The structures were optimized with the convergence criterion of 0.02 eV Å$^{-1}$. When calculating the density of states, an

electron convergence of $10^{-6}$ eV was used. A $1 \times 1 \times 1$ k-point mesh was used to sample the Brillouin zone. The Gibbs free energy for all reactions was obtained by the following formula Eq. 10:

$$\Delta G = \Delta E + \Delta ZPE - T\Delta S \tag{10}$$

Where $\Delta E$ represents the total energy change obtained from VASP. $\Delta ZPE$ and $\Delta S$ represent the zero-point energy change and entropy change, respectively. $T$ represents the reaction temperature, here is 298 K.

## Data availability
All data are available in the main text or the supplementary materials. Source data are provided with this paper.

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

## Acknowledgements

This work was supported by the National Natural Science Foundation of China (Nos. 11975122, 21906083, 22006067, 21908092, 12004180), Scientific and Technological Innovation Special Fund for Carbon Peak and Carbon Neutrality of Jiangsu Province (No. BK20220026), the China Postdoctoral Science Foundation (No. 2022M711614), and Jiangsu Funding Program for Excellent Postdoctoral Talent (Nos. 2022ZB194, 2022ZB197). We Acknowledge Guangai Sun and Yuanhua Xia (Key Laboratory of Neutron Physics and Institute of Nuclear Physics and Chemistry, China Academy of Engineering Physics, Mianyang, 621999, PR China) for kind assistance in processing the neutron diffraction data.

## Author contributions

J.M.: Conceptualization, funding acquisition, supervision, writing—review and editing. M.L.Z.: Supervision, writing—review and editing. M.M.: Writing—review and editing. C.J.H., Z.W.J., and Q.Y.W.: Investigation, validation, writing—original draft. S.Y.C. and Q.H.L.: Formal analysis. C.C. and Y.L.W.: Funding acquisition, writing—review and editing. W.Y.Y., J.B.Y. and L.Y.Y.: Methodology, writing—original draft. J.P. and W.Q.S.: Writing—review and editing.

## Competing interests

The authors declare no competing interests.
