## [Peer Review File · Nature Communications]

Selective CO₂ reduction to CH₃OH over atomic dual-metal sites embedded in a metal-organic framework with high-energy radiationREVIEWER COMMENTS

Reviewer #1 (Remarks to the Author):

The authors reported an unprecedented catalytic strategy to convert CO₂ into CH₃OH using a high-energy radiation technique combined with atomic engineering of MOFs-based catalysts, which differs from the existing thermochemical, electrolytic, and photolytic techniques. Overall, the authors made substantial efforts in this work, with insights into the high-energy radiation technique. However, the authors did not analyze some data in depth and there are some shortcomings.

1. The renewable high-energy radiation as the energy input for the chemical transformation of CO₂ and water to energy-rich fuels is interesting. Although the study is technically sound and well-performed, it offers insufficient novelty, meaning, and conceptual advancement. Moreover, how to consider the practicality?
2. The ⁶⁰Co γ -ray driven catalytic CO₂-to-CH₃OH performance of CuNi Nanoparticles/UiO-66(Hf) needs to be offered in the manuscript for comparison.
3. Is the high-energy radiation catalysis universal to other bimetallic materials? The high-energy radiation powering selective CO₂ reduction to CH₃OH over Cu-Ni dual-metal-sites embedded MOF should be compared with other dual-metal-sites systems. Comparative performance of heterogeneous catalysts under similar reaction conditions for CO₂-to-CH₃OH should be added into the manuscript.
4. To determine the stability of samples after ⁶⁰Co γ -ray driven catalysis, the Cycle test of 0.05 wt% CuNi SAs/UiO-66(Hf) in water with CO₂ saturated solution needs to be extended the number of cycles. In addition, more characterizations including TEM, EDS mapping, XPS etc. should be added.
5. In situ DRIFTS spectra of CO adsorption on CuNi SAs/UiO-66(Hf) as well as CuNi SAs/UiO-66(Hf) with CO₂ saturated solution under different γ -ray irradiation should be provided to exhibit the change of COOH* and CH₂O*, which is conducive to verify DFT calculations of CO₂RR.

Reviewer #2 (Remarks to the Author):

This paper presents exciting new results, demonstrating how CO₂ gas can be radiolytically reduced all the way to methanol with reasonable efficiency in a two-metal metal-oxide-framework. This may be of great practical importance in removing CO₂ from the atmosphere.

The g-value achieved is 0.15 micromoles/J, when scavenger for OH radicals is present. Without OH scavenger, of course, product methanol will just be re-oxidized by the OH radicals. Perhaps the authors could comment on how a practical industrial-scale process might incorporate a OH scavenger for reasonable energy efficiency (the OH scavenger has to be regenerated or discarded).

My only comment on the writing is in line 257 where it says .09 micromoles/J is more efficient than the hydrated electron yield in water. I did not understand this until the following paragraph, where it is made clear that six electrons are needed to generate methanol. Please revise the text to make it clear

immediately what is meant.

Otherwise, very fine work, congratulations!

Reviewer #3 (Remarks to the Author):

Review attached.

Reviewer #4 (Remarks to the Author):

The report by Hu et al. investigates the radiolytic assisted CO₂ conversion to CH₃OH using a bimetallic Cu-Ni MOF-based support. This is an interesting, well conducted study in an emerging research area. The authors implemented several in depth materials characterization techniques and theoretical investigations to support their findings. With that said, the novelty of this work is not as high as one would expect from the scope of this journal. Further, the concerns listed below limit the strength of the study, especially from a materials design perspective and its publication is not recommended in its current form.

1. First, it is already clear from the literature that the same parent MOF Hf-UiO-66 would be an appropriate support for this approach. Hf-UiO-66 was originally reported as a radio sensitizer in 2016 (Biomaterials 2016 Vol. 97 Pages 1-9).
2. Second, this group of authors already demonstrated the use of Zr/Hf-based nanoscale UiO-66 MOFs as highly effective and stable radiation sensitizers for purified and natural water splitting under γ -ray irradiation (<https://doi.org/10.1021/jacs.3c00547>), thereby associating two of the main design principles implemented in this study.
3. Third, the radiolytic assisted CO₂ conversion was pursued by this group, previously in a catalyst-free environment. (ChemPhysChem 22 (18), 1900-1906). As the authors acknowledge and cite, the use of single metal sites Cu@UiO-66 has been previously demonstrated for other catalytic applications.
4. In this context, the authors' claim that "The present study provides a unique and practical solution to tackle CO₂ emissions and energy storage." is not fully substantiated, since this approach is not unique and also necessarily practical in the current global landscape.
5. Overall, the authors are mainly focusing on providing convincing structure-function evaluations with a focus on the catalytic pathways but lack the same level of depth for the materials characterization. Additional supporting evidence of the atomic dispersion of the dual Cu-Ni metal sites, such as determining the location via diffraction techniques/Rietveld refinement and solid-state NMR would be highly beneficial to decipher and provide additional fundamental understanding that can be then further

applied for the design of superior catalytic materials.

6. There is no mention or characterization that demonstrates the Hf-UiO-66 parent material is indeed defective, and if is, to what extent. Further, it would be beneficial to provide a discussion on the much lower level of Ni vs Cu incorporation despite the excess introduced in the reaction. Is this the final result for an optimized composition? How do the reported results compare against the theoretical/calculated metal incorporation, and how does that further correlate with the number of defects in the parent Hf-UiO-66 MOF?

7. Also, the higher N₂ uptake and associated pore volume in the Cu-Ni functionalized material is not intuitive, so this aspect needs to be justified and discussed.

Point-by-point responses to the reviewers' comments

(Manuscript ID: NCOMMS-23-04135-T)

Review #1 :

The authors reported an unprecedented catalytic strategy to convert CO₂ into CH₃OH using a high-energy radiation technique combined with atomic engineering of MOFs-based catalysts, which differs from the existing thermochemical, electrolytic, and photolytic techniques. Overall, the authors made substantial efforts in this work, with insights into the high-energy radiation technique. However, the authors did not analyze some data in depth and there are some shortcomings.

1. The renewable high-energy radiation as the energy input for the chemical transformation of CO₂ and water to energy-rich fuels is interesting. Although the study is technically sound and well-performed, it offers insufficient novelty, meaning, and conceptual advancement. Moreover, how to consider the practicality?

Response: We appreciated the interests the reviewer expressed, and also critical comments that improved the manuscript. Radiation chemists have been finding ways to tackle the environmental problems humans have created. Over the past few decades, it has been demonstrated that ionizing radiation such as electron beam (EB) and gamma-ray radiation technologies for flue gas/VOCs treatment (SO_x and NO_x removal), and wastewater purification can be effectively deployed to mitigate environmental degradation¹. The implementation of EB radiation technology has been experienced in pilot plants and several industrial plants. Because of these contributions, the International Atomic Energy Agency (IAEA) has highlighted the use of radiation technology for environmental remediation as a key component in the peaceful use of nuclear technology and implemented coordinated research projects related to various aspects of this technology². So, radiation chemistry, a largely unrecognized branch of chemistry, has had far-reaching effects.

Regarding radiolytic CO₂ conversion, initial attempts were made during the 1970s-1990s³⁻⁶, which involved the direct decomposition of gaseous CO₂ to CO upon irradiation. To avoid the recombination and back reactions of CO₂ radiolysis, the mixture with saturated hydrocarbons such as CH₄, C₂H₆, C₃H₈, and C₄H₁₀ was suggested^{7,8}. Besides, the effect of γ -rays and metal ion additives has been studied using CO₂-saturated solutions suspended with iron powder or zeolite^{3,4}. While irradiation pathways facilitate conversion at room temperatures due to the formation of reactive intermediates, the conventional radiolytic pathways suffered from the side reactions, resulting in low radical yields (*G*-values, the numbers of molecules per 100 eV absorbed) and poor selectivity for specific products, in which CO is unavoidable in certain fraction. Besides, the CO₂ transformation insights at irradiated interfaces were lacking. Due to these obstacles, the strategy of radiation-induced CO₂ conversion was indeed once considered questionable for practical applications.

Now, we have come up with fundamental new chemistry that offers a creative solution to the challenge of recycling CO₂ into energy-rich CH₃OH with water. This work found the combination of high-energy radiation with contemporary catalysts, i.e., CuNi SAs/UIO-66(Hf), can overcome

the long-standing challenges of selectivity and energy efficiency. To the best of our knowledge, there are no reports so far on the selective and effective production of CH₃OH or any liquid fuels based on the high-energy system. Importantly, we further used pulse radiolysis, along with other experimental and theoretical methods, to elucidate the fundamental radiolytic steps of CO₂ activation, intermediates binding, hydrogenation, and eventual CH₃OH production. As some reviewers approved, this study represents an exciting conceptual advancement, offering new opportunities for CO₂ conversion beyond the existing thermal, photolytic, and electrolytic approaches, as well as beyond traditional radiolytic thinking.

We consider the practicality of our method according to the following four aspects:

Firstly, we demonstrate exceptional conversion efficiency and selectivity of CH₃OH under γ -ray irradiation with 0.05 wt% CuNi SAs/UiO-66(Hf). The *G* value of CH₃OH production was promoted to 0.9×10^{-7} mol/J. Because CO₂-to-CH₃OH conversion proceeds with a six-electron transfer, the measured *G* value of CH₃OH is two times higher $G(e_{aq}^-) \sim 2.8 \times 10^{-7}$ mol/J in neat water radiolysis. As an example, with an absorbed dose of 10 kGy, 500 mg catalyst can produce 0.9 mmol CH₃OH, and the trace of other products, such as CH₄, C₂H₄, and formic acid, is below the detection limit of the chromatography.

Secondly, the recycling performance of catalysts has been an essential practicability consideration. We have performed more measurement cycles to verify the stability and indicated that this specific catalytic performance is maintained even under 8 cycles of irradiation treatment. After the stability test, we conducted TEM, EDS mapping, and XPS on the samples to investigate whether any structural transformations occurred with continuous irradiation. The characterizations showed that structures of supported single atoms remain preserved, further confirming the durable radiation catalytic property of CuNi SAs/UiO-66(Hf).

Besides, practical considerations also should include careful technical-economic analysis. To avoid the re-oxidization of \bullet OH radicals, we used sulfite ions as a scavenger. Na₂SO₃ is a cost-effective chemical and could be readily produced through the cases of flu gas SO₂ removal process on an industry or pilot scale. The SO₂ removal efficiency can exceed 80% under optimized operating conditions including pH value, liquid-gas ratio, inlet SO₂ concentration, and initial Na₂SO₃ molar concentration. It is worth noting that industrial coal desulfurization produces approximately 10⁸ tons of sulfite industrial products every year in China, which have a very low lysogenic rate and pose a risk of secondary SO₂ release. By utilizing Na₂SO₃ as an effective \bullet OH scavenger, our research may provide a direct source of industrial treatment for CO₂ and SO₂, the major component of industrial exhaust gas.

At last, the large-scale practicality was verified by our supplementary experiment on the industrial electron accelerator with high-dose rate. There are currently over 1700 EB units in commercial use and these numbers are dramatically increasing year by year worldwide. The safety regulations have been well-established. So, high-current electron accelerators and ⁶⁰Co-gamma sources are used in diverse industries to enhance the physical and chemical properties of materials and to reduce undesirable contaminants such as pathogens, and toxic byproducts. For example, it

was reported in 2020 that the industrial wastewater treatment facility using EB (electron beam) technology showed the capacity to treat 30,000 cubic meters of wastewater per day^{2,9}. In this regard, an EB was selected to evaluate the CH₃OH generation efficiency of CuNi SAs/UiO-66(Hf). Since EB output a large amount of energy in a short time and convert sustainable electrical energy into radiation energy with high conversion efficiency (up to 80%), resulting in an impressive yield in our experiment. Specifically, we achieved a remarkable rate of 0.23 mmol/(g·min) with an irradiation time of only 2 minutes and an absorbed dose of 80 kGy using 50 mg of catalyst in a 50 mL reaction solution. As a result, we obtained a CH₃OH concentration of 0.55 mmol/L. The production rate has exceeded the majority of photocatalytic and electrocatalytic synthesis efficiencies, which highlights the exceptional performance of high-energy radiation-based CO₂-to-CH₃OH conversion.

Overall, our work not only proposed the new conceptual radiolytic catalysis but also offers practical solutions for CO₂ reduction and the utilization of greenhouse gases. The selective production of CH₃OH and the integration of radiation and catalysis hold great promise for addressing global challenges related to energy, the environment, and carbon management.

2. The ⁶⁰Co γ -ray driven catalytic CO₂-to-CH₃OH performance of CuNi Nanoparticles/UiO-66(Hf) needs to be offered in the manuscript for comparison.

Response: We appreciate with the comments. We further synthesized CuNi nanoparticles (NPs) and UiO-66(Hf) supported CuNi NPs composites based on the reported solvent-thermal methods. Their performance in CH₃OH production was conducted at identical irradiated conditions, and the results are shown in Figure 1. With dispersed CuNi NPs alone, we found that the radiolytic approach still can produce CH₃OH; however, both the total yield and selectivity were far lower than the values with CuNi NPs/UiO-66(Hf) and CuNi SAs/UiO-66(Hf). On the other hand, CuNi NPs/UiO-66(Hf) displays a comparable selectivity, while the efficacy is only about 60% that of CuNi SAs/UiO-66(Hf).

Figure 1. Catalytic activities of CuNi nanoparticles (NP), CuNi NP/UIO-66 in comparison with CuNi SAs/UIO-66.

These supplementary data have several implications, which could support the assertions made and conclusions drawn in the work.

First, once CO_2 radical anions ($\text{CO}_2^{\cdot-}$) are produced in solutions, CuNi NPs even in the absence of MOFs can act as active sites to adsorb some of the species, and make the subsequent e^-/H^+ transfer possible. The rest of $\text{CO}_2^{\cdot-}$, however, undergoes competitive disproportion to generate CO. So, the selectivity for CH_3OH is limited to 60%. The observation agrees well with the transient kinetics based on pulse radiolysis experiments.

Second, the higher efficacy with UiO-66(Hf) matrix can be attributed to its dual role of “sensitization”. In accordance with previous reports of radiation therapy and radiolytic water splitting^{10–13}, UiO-66(Hf) exhibits an enhanced scattering cross-section for Auger/Compton electrons at the nanoscale interface, which significantly increases the availability of reactive electrons. In addition, the unique topology likely creates a confined environment to accelerate CO_2 reduction process. In other words, catalytic reactions taking place within the nanopores of UiO-66(Hf) materials with dimensions below 1 nm experience strong confinement effects which impact the reactivity.

Third, we found atomic CuNi catalysts are superior over NPs. Especially, for metal NPs like Cu, their size-dependent electronic structures will be reflected on their catalytic behavior in radiation catalysis. In the case of supported single atoms, they can be stabilized by the support through chemical bonding, especially when single atoms are anchored on MOFs supports or transition metal oxides and zeolites. Thus, those single atoms may show limited geometric transformation under reaction conditions. In the case of metal nanoparticles (>1 nm, usually with more than 40 atoms), their geometric structures are less sensitive, and usually the geometric structure of one metal nanoparticle is relatively stable, although the geometric configuration of exposed surface atoms (facet, corner, edge, metal–support interface, etc.) may change due to the environment.

3. Is the high-energy radiation catalysis universal to other bimetallic materials? The high-energy radiation powering selective CO_2 reduction to CH_3OH over Cu-Ni dual-metal-sites embedded MOF should be compared with other dual-metal-sites systems. Comparative performance of heterogeneous catalysts under similar reaction conditions for CO_2 -to- CH_3OH should be added into the manuscript.

Response: thanks again for this comment. To address this important question, we have synthesized various dual-metal-sites embedded UiO-66(Hf) i.e., Cu/Co, Cu/Mn, Ni/Co, Ni/Mn. The catalytic activities in Figure 2, showed that other bimetallic materials cannot grant comparable performance during the high-energy radiation catalysis. However, the high selectivity observed in series of Cu/Ni, Cu/Co, Cu/Mn suggested the important role of metallic copper.

Our early and ongoing pulse radiolysis results found that the metallic Cu substantially extended

the lifetime of $\text{CO}_2^{\cdot-}$ radicals by at least 10 times compared to them on Ni surfaces through the initial surface stabilization process, laying the groundwork for the subsequent multi-electron transfer reaction. For instance, the differentiation in the transient kinetics of $\text{CO}_2^{\cdot-}$ radicals adsorbed on Cu and Ni surfaces suggest diverse stabilization behavior and adsorbed structures of $\text{CO}_2^{\cdot-}$ radicals, which determines the subsequent selective reduction pathway of $\text{CO}_2^{\cdot-}$ radicals (Figure 3).

Figure 2. The catalytic performance of various dual-metal-sites embedded UiO-66(Hf) i.e., Cu/Co, Cu/Mn, Ni/Co, Ni/Mn.

Figure 3. a-d. Transient kinetics at 350 nm within 700 ns (a), 3D stereographs of transient absorption spectra within 7 μs (b), transient kinetics at 450 nm (c), and transient absorption matrix (d) within 80 μs in the presence of Cu NP. e-h, Transient kinetics at 350 nm within 700 ns (e), 3D stereograph of transient absorption spectra within 7 μs (f), transient kinetics at 350 nm (g), and transient absorption matrix (h) within 80 μs in the presence of Ni NP. (Unpublished data)

4. To determine the stability of samples after ^{60}Co γ -ray driven catalysis, the cycle test of 0.05 wt% CuNi SAs/UiO-66(Hf) in water with CO_2 saturated solution needs to be extended the number of cycles. In addition, more characterizations including TEM, EDS mapping, XPS etc. should be added.

Response: The durable radiation-resistance of catalysts has been an essential practicability consideration. We have performed more measurement cycles to verify the stability. The Figure 4 demonstrated that this specific catalytic performance is maintained even under 8 cycles of irradiation treatment. After the stability test, we conducted TEM, EDS mapping, and XPS on the samples to investigate whether any structural transformations occurred with continuous irradiation. The characterizations showed that structures of supported singles atoms remain preserved, further confirming the durable radiation-resistance of CuNi SAs/UiO-66(Hf). However, as the reaction progresses, the protic solvent water can leach Ni atoms from UiO-66(Hf) over several cycles. The ICP results indicate that the content of Ni element is reduced to 0.015% after eight cycles, which also explains the absence of Ni XPS detection after cycling. This phenomenon may be one of the factors contributing to the degradation in the performance of CO₂ to CH₃OH.

Figure 4 Cycle test of 0.05 wt% CuNi SAs/UiO-66(Hf) in water with CO₂-saturated solution under 1-4 kGy γ -ray irradiation. TEM EDS mapping and XPS of CuNi SAs/UiO-66(Hf) before and after 8 times stability test.

5. In situ DRIFTS spectra of CO adsorption on CuNi SAs/UiO-66(Hf) as well as CuNi SAs/UiO-66(Hf) with CO₂ saturated solution under different γ -ray irradiation should be provided to exhibit the change of COOH* and CH₂O*, which is conducive to verify DFT calculations of CO₂RR

Response: Thanks for this suggestion. The evolution of hydrogenated intermediates like COOH* and CH₂O* indeed is critical to the overall CH₃OH production. We conduct DRIFTS spectral measurements under deep UV irradiation (193~248 nm). The results in Figure 5, clearly showed the formation of surface-bound COOH* and CH₂O* within minutes. The transient peaks of these intermediates are in the consistence with previous reports¹⁴. Fortunately, solvated electrons could be formed through the excitation by UV light in Na₂SO₃ solution¹⁴, and the band gap of UiO-66 was

also in line with the energy of deep UV light to produce the photoelectron for $6\text{H}^+/6\text{e}^-$ reduction of CO_2 -to- CH_3OH . As shown in Figure 6a, the peak of CO_2^* and HCO_3^* as a carbon dioxide adsorption species was located 1692 cm^{-1} and 1444 cm^{-1} , which may act as active bodies on the catalyst surface to provide a source of CH_3OH . The COOH^* absorption band at 1558 cm^{-1} and 1648 cm^{-1} gradually increased along with the increase in illumination time, revealing the hydrogenation of CO_2^* radicals or CO_2 on single atom sites. Furthermore, the formation of CH_3O^* was suggested by the stretching vibration band at 1127 cm^{-1} , which represents the crucial intermediates that are directly related to the CO_2 to CH_3OH . It should be noted that an in-situ DRIFTS setup under γ -ray irradiation has not been developed so far. The highly penetrating γ -ray irradiation often causes the malfunction of electronic devices, making it almost impossible to position the instrument close to the ^{60}Co γ -ray sources. Instead, we are currently updating a custom-designed DRIFTS setup coupled with X-rays. We anticipate that these ongoing efforts will provide further insights into radiolytic CO_2 reduction and relevant radiation catalysis process.

Figure 5 In-situ DRIFTS spectra for detecting the reaction intermediates in CO_2 reduction over CuNi SAs/UiO-66(Hf) under deep UV irradiation.

Reference

1. Trojanowicz, M., Bobrowski, K., Szreder, T. & Bojanowska-Czajka, A. Gamma-ray, X-ray and electron beam based processes. in *Advanced oxidation processes for wastewater treatment: emerging green chemical technology* 257–331 (Elsevier Inc., 2018).
2. IAEA. Radiation treatment of polluted water and wastewater. *IAEA-Tecdoc* (2008).
3. Fujita, N. & Matsuura, C. Radiation induced reduction of CO_2 in iron containing solution. *Radiation Physics and Chemistry* **43**, 205-213 (1994).

4. Fujita, N., Morita, H., Matsuura, C. & Hiroishi, D. Radiation induced CO₂ reduction in an aqueous medium suspended with iron powder. *Radiation Physics and Chemistry* **44**, 349-357 (1994).
5. Ershov, B. G., Janata, E., Michaelis, M. & Henglein, A. Reduction of Cu²⁺_(aq) by CO₂⁻: First steps and the formation of colloidal copper. *Journal of Physical Chemistry* **95**, 8996-8999 (1991).
6. Wu, X. Z., Hatashita, M., Enokido, Y. & Kakihana, H. Reduction of carbon dioxide in γ ray irradiated carbon dioxide: Water system containing Cu²⁺ and SO₃²⁻. *Chem Lett* **29**, 572-573 (2000).
7. Ikezoe, Y. & Sato, S. Radiation chemical reactions in carbon dioxide-propane system formation of carbon monoxide by fission fragments. *J Nucl Sci Technol* **13**, 503-507 (1976).
8. Dyer, A. & Moore, G. E. The radiolysis of simple gas mixtures-I. Rates of production and destruction of methane in mixtures with carbon dioxide as a major constituent. *Radiation Physics and Chemistry* **20**, 315-321 (1982).
9. Pearce, R., Li, X., Vennekate, J., Ciovati, G. & Bott, C. Electron beam treatment for the removal of 1,4-dioxane in water and wastewater. *Water Sci Technol* **87**, 275-283 (2023).
10. Xu, Z. *et al.* Monte Carlo simulation-guided design of a thorium-based metal-organic framework for efficient radiotherapy-radiodynamic therapy. *Angew. Chem. Inter. Ed.* **61**, e202208685 (2022).
11. Lan, G., Ni, K., Veroneau, S. S., Song, Y. & Lin, W. nanoscale metal-organic layers for radiotherapy-radiodynamic therapy. *J Am Chem Soc* **140**, 16971-16975 (2018).
12. Ni, K. *et al.* Nanoscale metal-organic frameworks for mitochondria-targeted radiotherapy-radiodynamic therapy. *Nat Commun* **9**, 4321 (2018).
13. Liu, J. *et al.* Nanoscale metal-organic frameworks for combined photodynamic & radiation therapy in cancer treatment. *Biomaterials* **97**, 1-9 (2016).
14. Liu, Z. *et al.* Prebiotic photoredox synthesis from carbon dioxide and sulfite. *Nat Chem* **13**, 1126-1132 (2021).

Review #2 :

This paper presents exciting new results, demonstrating how CO₂ gas can be radiolytically reduced all the way to methanol with reasonable efficiency in a two-metal metal-oxide-framework. This may be of great practical importance in removing CO₂ from the atmosphere.

Response: Thanks very much for the comments. The g-value achieved is 0.15 micromoles/J, when scavenger for OH radicals is present. Without OH scavenger, of course, product methanol will just be re-oxidized by the OH radicals. Perhaps the authors could comment on how a practical industrial-scale process might incorporate a OH scavenger for reasonable energy efficiency (the OH scavenger has to be regenerated or discarded).

Response: We greatly appreciate this comment. Among many OH scavengers, our CO₂ reduction work found the best activity with Na₂SO₃. Sulfite is a cost-effective chemical and could be readily produced through the flue gas SO₂ removal process on an industry or pilot-scale. The SO₂ removal efficiency can exceed 80% under optimized operating conditions including pH value, liquid-gas ratio, inlet SO₂ concentration, and initial Na₂SO₃ molar concentration.

Industrial coal desulfurization produces approximately 10⁸ tons of sulfite every year in China,

which have a very low lysogenic rate and pose a risk of secondary SO₂ release¹. By utilizing Na₂SO₃ as an effective OH scavenger, our research may provide an integrated treatment for CO₂ and SO₂, the major component of industrial exhaust gas.

My only comment on the writing is in line 257 where it says 0.9 micromoles/J is more efficient than the hydrated electron yield in water. I did not understand this until the following paragraph, where it is made clear that six electrons are needed to generate methanol. Please revise the text to make it clear immediately what is meant.

Response: Thanks for point out this issue. We have made it clearly in the revised manuscript.

Otherwise, very fine work, congratulations!

Reference

1. Liu, S., Liu, W., Jiao, F., Qin, W. & Yang, C. Production and resource utilization of flue gas desulfurized gypsum in China - A review. *Environ. Pollut.* **288**, 117799, (2021).

Review #3 :

The authors report the use of a hafnium (Hf)-based metal organic framework (MOF) that contains dual metal (Cu/Ni) single atom sites for the efficient and selective conversion of CO₂ to CH₃OH in aqueous solution under the application of ionizing radiation (gamma rays). The 3D array of high-Z Hf nodes acts as a radiation sensitizer, producing a cascade of low energy electrons that enhances the yield of hydrated electrons compared to in the radiolysis of bulk water. The CO₂ conversion is initiated by the reduction of CO₂ to CO₂^{•-} by the hydrated electrons, and then the single atom Ni and Cu sites serve to catalyze the 6- electron/6-proton reduction of CO₂ to CH₃OH via a series of interfacial reactions involving adsorbed carbon species, starting from adsorbed CO₂^{•-}. Another advantage of using a MOF is its huge specific surface area and its porosity towards CO₂, allowing CO₂ to be easily captured and channeled into the MOF framework. I believe that this work represents an exciting breakthrough that could potentially offer a novel way to use abundant high-energy radiation, that is readily available at nuclear reactors and/or electron accelerators, to convert CO₂ into an energy-rich, liquid fuel. I would therefore recommend publication after the following points have been addressed:

Response: We appreciated with these encouraging comments and revised the manuscript accordingly.

1. A general comment is that no real indication is given in the manuscript as to how much methanol we could expect to generate in this process, for example per gram of MOF after a certain amount of irradiation time with a particular radiation source. Very nice data are presented in terms of G-values of the products, but I was left with no feeling of the quantity of methanol that can be generated. For example, are we talking about a few microliters of methanol per gram of MOF, or is it more than that?

Response: Thanks for this importance comment. Our earlier attempt showed that the use of a small quantity (<80 mmol/L) of UiO-66(Hf)-OH can achieve a γ -rays-to-hydrogen conversion efficiency

exceeding 10% that significantly outperforms Zr-/Hf-oxide nanoparticles and the existing radiolytic H₂ promoters. In UiO-66(Hf), the combination of 3D arrays of ultras-small metal-oxo clusters and high porosity affords unprecedented effective scattering between secondary electrons and confined water, generating increased precursors of solvated electrons and excited states of water, which are the main species responsible for H₂ production enhancement. In this work, the irradiation cycle test was carried out with 0.05 wt% CuNi SAs/UiO-66(Hf) in CO₂ saturated aqueous solution under 4-32 kGy γ -ray irradiation. So, it is estimated, for instance with absorbed dose of 10 kGy, 0.5 gram catalysts in 1 liter water would produce 0.9 mmol (28.8 mg) CH₃OH. It is noted that the absorbed dose per minute (dose rate) mainly relies on radioisotope Co-60 activity or electron beam current. Typically, the dose rate of an electron beam varies from 40 kGy/min up to 100 kGy/s. In this case, one would expect the feasibility of CH₃OH production via our newly-developed approach.

Figure 6. The production of CH₃OH with 0.01 wt% catalyst CuNi SAs/UiO-66(Hf) in .01 M Na₂SO₃ solution with CO₂ saturated under 0-80 kGy electron beam at 200 keV.

In this regard, electron accelerator, a well-established high-energy system utilized in numerous industrial applications for the desulfurization and denitrification of waste gas, was selected to evaluate the CH₃OH generation efficiency of CuNi SAs/UiO-66(Hf). Since electron beams output a large amount of energy in a short time and convert sustainable electrical energy into radiation energy with high conversion efficiency (up to 80%), resulting in impressive yield in our experiment (Figure 6). Specifically, we achieved a remarkable rate of 0.27 mmol/(g·min) with an irradiation time of only 2 minute and an absorbed dose of 80 kGy using 1g of catalyst in a 1L reaction solution.

2. I am not an expert on MOF structures, and it took me a little time to realize that the Hf nodes are the secondary building units (SBUs). I would therefore suggest a small change in wording in lines 97-101, to the following: “These materials feature exceptionally high specific areas and secondary building units (SBUs) composed of high-Z metal nodes such as Hf and Zr, which can be easily modified to display an extremely high probability of interacting with incoming ionizing radiation. The 3D arrays of nanoscale SBUs greatly enhance the scattering.....”

Response: we have revised the wording when we introduce the materials.

3. Fig. 2: In panel a of Fig 2, I strongly recommend that you add “85 °C” next to or underneath “Fixation”.

Response: thanks, it would indeed be clearer when we add to the synthesis process. we have revised the Fig.2.

4. Line 207: I assume you meant to write “EXAFS”, not “EXANFS”?

Response: we have revised the typing mistake.

5. Line 217: I assume to meant to write “XANES”, not “XANFS”?

Response: we have revised the typing mistake.

6. Line 258: Saying here that the G value of CH₃OH of 0.9×10^{-7} mol/J “far exceeds the eq- values in pure water” is confusing since the G value of eq- in pure water is 2.8×10^{-7} mol/J.

Response: it is made clear that six electrons are needed to generate methanol.

7. Line 272: Change “SUB” to “SBU”.

Response: We have revised this error.

8. Fig. 5: Please add in the caption what mass fraction of the MOF was used in the pulse radiolysis experiments.

Response: We have added the mass fraction of MOF in the caption. We note the transparency of the solutions at such loading percentage is compatible with transient absorption spectroscopy.

9. Fig. 5: The caption says that panel c is at 360 nm and panel d is at 520 nm. Please double-check these wavelengths. I feel like you might have got these wavelengths mixed up.

Response: Thank you for your careful review, we got these two mixed up and have revised them.

10. Fig. 5: I would add some labels onto Fig. 5 (panels c and d) to make it much easier for the reader to interpret what is being shown here. If I am correct in my understanding of these figures, in panel c, I would add “e_{aq}⁻” at the top of the initial fast rise, and “MOF-e_{aq}⁻” above the residual absorption at longer time delays. In panel d, I would add “CO₂⁻”.

Response: We have added these essential labels onto Fig. 5 and we think this makes it more intuitive to understand.

11. Lines 319/320: I would insert a few words here that explain to non-radiation chemists why tert-butanol was used. For example, “.....in Ar-saturated 0.1 M tert-butanol solutions, in which no CO₂⁻ is formed (Fig. S16),.....”.

Response: We have revised this sentence to: “Compared with the reference spectrum measured in Ar-saturated 0.1 M tert-butanol solutions, in which no CO₂⁻ formed and only e_{aq}⁻ remained since

the scavenging of $\cdot\text{OH}$ and $\cdot\text{H}$ radical by tert-butanol (**Fig. S18**), the peak at 520 nm is likely assigned to adducts complex formed via the reaction between UiO-66 and e_{aq}^- .”

12. Lines 326-329: It is mentioned here that “the typical transient absorption spectrum of $\text{CO}_2^{\cdot-}$ radical was also readily identified in the 300-400 nm region.” However, $\text{CO}_2^{\cdot-}$ in water absorbs at 235 nm and it has practically no absorbance in the 300-400 nm region (see P. Neta et al. J. Phys. Chem. 1969, 73, 4207-4213). Are the authors suggesting that free $\text{CO}_2^{\cdot-}$ is not observed, and that only the (Cu/Ni)M- $\text{CO}_2^{\cdot-}$ adduct is formed, which exhibits a spectrum that is significantly red-shifted relative to free $\text{CO}_2^{\cdot-}$ in water? This might be difficult to believe though, considering that the samples are dispersions of a very small amount of MOF in water, so a very large fraction of the signal must be coming from reactions in the bulk water, outside of the MOF.

Response: We have performed pulse radiolysis experiments in CO_2 -saturated formate solution without MOF. Within tens of nanoseconds, e_{aq}^- decays rapidly and follows a new absorption spectrum raised, corresponding to $\text{CO}_2^{\cdot-}$ radicals in the solution. In the report of P. Neta, they also observed little absorption of $\text{CO}_2^{\cdot-}$ above 300 nm, but indeed a very low absorption coefficient ($\epsilon < 500$).

Figure 7. 3D stereograph of transient absorption spectra at different time with 0.1 M formate.

13. Line 369: Define “TDOS”.

Response: Thank you for the careful review, we got these two mixed up and have revised them.

14. Line 390: Define “PDOS”.

Response: Thank you for your comment, we got these two mixed up and have revised them.

15. Line 391: Presumably you meant to write “hydrogenation”, not “dehydrogenation”? 16. Supporting Information, Table S1: In the title of this table, change “Cu and loadings” to “Cu and Ni loadings”.

Response: Thank you for your careful review, we got these two mixed up and have revised them.

Review #4 :

The report by Hu et al. investigates the radiolytic assisted CO₂ conversion to CH₃OH using a bimetallic Cu-Ni MOF-based support. This is an interesting, well conducted study in an emerging research area. The authors implemented several in depth materials characterization techniques and theoretical investigations to support their findings. With that said, the novelty of this work is not as high as one would expect from the scope of this journal. Further, the concerns listed below limit the strength of the study, especially from a materials design perspective and its publication is not recommended in its current form.

Response: We appreciated the comments on materials design and characterizations. After years of effort, we are currently pioneering an emerging concept, radiolytic catalysis, which is rooted in basic radiation chemistry, aiming to solve the most pressing environmental and energy challenges, including water splitting to hydrogen, CO₂ reduction, ammonia synthesis, and so on. Radiolytic catalysis is the integration of ionizing radiation and contemporary catalysts to achieve reactant conversions and product selectivity that are inaccessible to radiation or catalysts alone. While chemical transformations via radiation and catalysis are individually well-developed and optimized in many cases, efficient and effective radiolytic catalysis coupling remains primitive. Importantly, time-resolved and atomic understanding of radiolytic catalysis is further complicated by the intricate natures of radiation effects and catalysis separately.

Against this backdrop, we recently demonstrated effective water splitting assisted by Zr/Hf-based nanoscale UiO-66 MOFs upon γ -rays irradiation. The approach achieved a remarkable rays-to-hydrogen efficiency exceeding 10%, outperforming the existing radiolytic H₂ promoters.

In addition to H₂ production, CO₂-to-CH₃OH conversion represents one of the most challenging processes because it involves six electron/H transfer. To address this challenge, we have developed a novel intermetallic catalyst synthesized via a facile radiolytic reduction process, combining copper (Cu) and nickel (Ni) at atomic level. In order to highlight these novelty and significance of our work, we will provide additional elaboration in the Introduction section. This will ensure that readers can quickly grasp the innovative aspects and the potential impact of our research in the field of radiolytic catalysis.

1. First, it is already clear from the literature that the same parent MOF Hf-UiO-66 would be an appropriate support for this approach. Hf-UiO-66 was originally reported as a radio sensitizer in 2016 (Biomaterials 2016 Vol. 97 Pages 1-9).

Response: We appreciate that the reviewer pointed out the first report on the radiation sensitization role of MOF UiO-66(Hf). The sensitization ability of MOF upon X-ray irradiation produces very reactive molecules which don't travel far from the injection site—they latch on and stay right where you put them. In this case, the frameworks absorb radiation better than tissue, delivering an extra dose of radiation to the tumor. The oxidizing \cdot OH radicals and O₂⁻ are mainly responsible for the radiation therapy. However, this work uncovered that secondary electron scattering in confined environment play a significant role, which is rarely reported in the existing literature. In addition, the

use of MOFs materials has never been reported on radiolytic CO₂ reduction.

2. Second, this group of authors already demonstrated the use of Zr/Hf-based nanoscale UiO-66 MOFs as highly effective and stable radiation sensitizers for purified and natural water splitting under γ -ray irradiation (<https://doi.org/10.1021/jacs.3c00547>), thereby associating two of the main design principles implemented in this study.

Response: As mentioned by the reviewer, we reported on MOF amplified radiolytic H₂ production. The results unraveled the sensitization mechanism of MOF in the aqueous system. We found the increased radiolytic yield of excited-state water molecules and precursors of solvated electrons due to the enhanced scattering with SBU. However, since H₂ is already one of the primary molecular products of water radiolysis, our earlier work did not alter the type of radiolytic products but only increased the radiolytic yield (G value) of H₂, which is within the scope of conventional radiation chemistry of water.

Almost at the meantime, we made extensive attempts in catalytic CO₂ reduction because the reduction of CO₂ differs significantly from H₂ generation. The primary products of CO₂ aqueous radiolysis are carbon monoxide and oxalic acid, involving two-electron transfer reactions. It is challenging to form alcohols and alkanes through multiply electron/H transfer, a deeper conversion. However, this study leverage the radiation sensitization effect of MOF but also, more importantly, employed MOF as matrix/support for bimetallic single-atoms to selectively regulate the types of radiolysis products, especially CH₃OH. These resulted in the generation of entirely new products distinct from those produced in traditional radiolysis processes.

This work breaks the fundamental limitations of common radiation products and achieves the selective transformation of transient free radical intermediates in high-energy radiation processes. It also opens up new possibilities for other fules beyond traditional radiolysis processes and shows practical significance in expanding scope. The integration of radiation and catalytic processes therefore represents a significant advancement, offering unique opportunities for selective and controlled CO₂ transformations.

3. Third, the radiolytic assisted CO₂ conversion was pursued by this group, previously in a catalyst-free environment. (ChemPhysChem 22 (18), 1900-1906). As the authors acknowledge and cite, the use of single metal sites Cu@UiO-66 has been previously demonstrated for other catalytic applications.

Response: Our previous report primarily focused on the reactions and product analysis of catalyst-free CO₂ radiolytic reduction. Although conducted in a catalyst-free environment, the system added formate ions as reactants in addition to CO₂. In contrast, the current study exclusively utilized CO₂ as the only carbon source, and achieve CO₂ reduction to CH₃OH.

As mentioned above, the products obtained in the preliminary work mainly consisted of oxalic acid and carbon monoxide, which are typical two-electron products commonly observed in conventional radiation chemistry. We did not discover novel reaction processes overcoming the limitation until this work. However, in the present study, by incorporating of atomic Cu-Ni dual-metal-sites embedded MOF, we have achieved the selective production of a six-electron product,

methanol, via radiation reduction of CO₂. Compared to oxalic acid, this finding holds new promises. Cu@UiO-66 was used in photolytic or electrolytic process, whereas it has not been investigated in distinct radiolytic system. Furthermore, we conducted additional experiments involving the loading of different metal single atoms and nanoparticles onto the MOF. The results confirmed the unique catalytic activity of the atomic copper-nickel bimetallic system, consistent with our pulse radiolysis experiments. These additional experiments further support the efficacy and potential of our proposed catalyst system.

4. In this context, the authors' claim that "The present study provides a unique and practical solution to tackle CO₂ emissions and energy storage." is not fully substantiated, since this approach is not unique and also necessarily practical in the current global landscape.

Response: We have supplemented data and used industrial electron beam to strengthen the practicality of our approach. Additional illustrations were provided in revised manuscript to demonstrate the novelty.

Firstly, we have successfully demonstrated the selective reduction of CO₂ to CH₃OH using a bimetallic Cu-Ni catalyst embedded in a metal-organic framework (MOF). This achievement is remarkable as it involves the conversion of a six-electron transfer process, surpassing the typical two-electron reactions observed in conventional radiation chemistry.

Secondly, our study combines the synergistic effects of radiation and catalysis, enabling reactant conversions and product selectivity that are often inaccessible in radiation or catalysts alone. By leveraging the unique properties of the Cu-Ni SAs/UiO-66(Hf) catalyst, we have achieved efficient radiolytic reduction of CO₂ to CH₃OH. This integration of radiation and catalytic processes expands the application scope and practical significance of radiation chemistry.

Furthermore, our work has practical implications for industrial applications. We have utilized electron beams as an energy input, which are already widely employed in industries for waste gas desulfurization and denitrification. By harnessing electron beams for CO₂ reduction, we can leverage existing infrastructure and technology, making the transition to large-scale implementation more feasible. The high conversion efficiency of electron beams, coupled with the impressive yield of methanol achieved in our study, surpasses the efficiency of many conventional photocatalytic and electrocatalytic synthesis methods. This highlights the potential for electron beam-based radiolytic catalysis as a promising approach for efficient and sustainable CO₂ conversion.

Overall, our work not only advances the fundamental understanding of radiolytic catalysis but also offers practical solutions for CO₂ reduction and the utilization of greenhouse gases. The selective production of methanol and the integration of radiation and catalysis hold great promise for addressing global challenges related to energy, the environment, and carbon management.

5. Overall, the authors are mainly focusing on providing convincing structure-function evaluations with a focus on the catalytic pathways but lack the same level of depth for the materials characterization. Additional supporting evidence of the atomic dispersion of the dual Cu-Ni metal

sites, such as determining the location via diffraction techniques/Rietveld refinement and solid-state NMR would be highly beneficial to decipher and provide additional fundamental understanding that can be then further applied for the design of superior catalytic materials.

Response: We appreciated with these comments, and performed additional ^1H -NMR and neutron analysis of the samples to reveal the design principle.

Figure 8. Dissolution/ ^1H NMR and neutron powder diffraction spectra of UiO-66(Hf) and CuNi SAs/UiO-66(Hf).

Previous studies reported two types of defects in UiO-66: the absence of metal clusters and the absence of ligands. While many studies have focused on ligand deletion in monatomic systems, it is noteworthy that oxygen atoms linked to SBU (Secondary Building Units) serve as monatomic ligand sites^{1,2}. Therefore, our study primarily focuses on investigating defective ligand deletion using liquid NMR spectroscopy as a starting point. Firstly, in the case of UiO-66(Hf) with ligand loss, oxygen atoms lacking ligands are likely supplied by two carboxylate ions (HCOO^- and CH_3COO^-)^{3,4}, one of which originates from formate ions produced through DMF pyrolysis. Interestingly, our UiO-66 synthesis process did not involve the addition of acetate (CH_3COO^-) as a regulator. However, the congruent presence of CH_3COO^- observed in our results aligns with those reported in the literature. Acetate may come from impurities such as solvents, ligands or metal salt during synthesis, which may serve as a source of coordinating ions for ligand deletion in the MOF. These pieces of evidence were corroborated by the NMR data.

Next, we proceeded to refine the neutron diffraction data of UiO-66 (Hf). It has been previously noted that X-ray data collection for UiO-66, which contains heavy metal atoms like Zr and Hf, exhibits high sensitivity towards these metal atoms but lacks sensitivity towards light atoms (C, H, O) in the absence of ligands⁵. We replaced the ligand with deuterated benzene-1,4-dicarboxylic acid to obtain better experimental data. Conversely, neutron diffraction provides sensitivity to the light atoms. Based on the collected powder neutron diffraction data, we initially assumed a fully saturated state for the atomic occupancy of elements in UiO-66(Hf) without any ligands. This means that the atomic occupancy was set to 100%, and we obtained the corresponding fitting parameters (goodness of fit data: $R_p = 1.24$, $R_{wp} = 1.58$, $\chi^2 = 7.82$). However, as we defined the center of gravity of the defect as a ligand defect, the Hf periphery of the center, composed of $[\text{Hf}_6\text{O}_4(\text{OH})_4]$, still exhibited 8

coordination oxygen atoms. We assumed that the missing oxygen atom, resulting from the absence of a partial dispenser, was provided by carboxylic acids (HCOO^- and CH_3COO^-). Hence, the atomic occupancy of Hf and O elements was assumed to be 100%. Our results revealed that the ligand-contributed atoms occupied approximately 92% of UiO-66(Hf) (goodness of fit data: $R_p = 1.23$, $R_{wp} = 1.57$, $\chi^2 = 7.71$), resulting in better fitting results compared to those obtained without considering defects. These findings indicate the presence of ligand defects in UiO-66.

6. There is no mention or characterization that demonstrates the Hf-UiO-66 parent material is indeed defective, and if is, to what extent. Further, it would be beneficial to provide a discussion on the much lower level of Ni vs Cu incorporation despite the excess introduced in the reaction. Is this the final result for an optimized composition? How do the reported results compare against the theoretical/calculated metal incorporation, and how does that further correlate with the number of defects in the parent Hf-UiO-66 MOF?

Figure 9. The content and catalytic activity of different elements given by ICP before and after radiation reduction (1M isopropanol).

Response: The material defects were characterized using NMR and neutron diffraction, and the results were as expected. Our report highlights that the solvothermal system used in this synthesis method is not purely regulated by formic or acetic acid defect regulators, and therefore defects are present. Despite this, we observed different activities of CO_2 conversion to CH_3OH caused by different Cu/Ni content ratios. During the catalysts synthesis process, we found that exposure of Cu and Ni precursor to radiation reduction in isopropyl alcohol aqueous solution caused a rapid decrease in the content of Ni elements. The Ni metal continued to decline after several cycles of radiation reduction of carbon dioxide in treated the CuNi SAs/UiO-66(Hf), which may lead to degradation of material properties after cycles test. The loss of Ni element after contact with protic solvent (H_2O) has been reported previously¹. Hence, while defects may provide coordination sites for Cu and Ni metals in the heat treatment stage, the Ni content is significantly reduced after

radiation treatment in aqueous solution, and the performance evolution brought by the metal content may not have a clear structure-activity relationship with the defect content.

7. Also, the higher N₂ uptake and associated pore volume in the Cu-Ni functionalized material is not intuitive, so this aspect needs to be justified and discussed.

Response: To collect more accurate data, we adjusted the sample quantity (700 mg) for BET adsorption and re-collected nitrogen adsorption data for both the UiO-66(Hf) carrier and the single atomic catalyst CuNi SAs/UiO-66(Hf) (Figure 10). Additionally, we added data on CO₂ adsorption. Surprisingly, we found that exposure to only 4kGy during the radiation reduction of single atoms increased the specific surface area of the support and catalyst, as well as their CO₂ adsorption performance. Our comparable study on the gas adsorption performance of HKUST-1 after irradiation yielded similar results (Figure 11). Irradiation can decompose coordination water molecules rich or other impurities in MOF, leading to increased N₂ and CO₂ adsorption and specific surface area. This phenomenon has been confirmed in earlier reports on MOF irradiation⁶. The presence of Cu and Ni single atoms may also enhance CO₂ adsorption performance, but the extent is limited due to the low loading percentage.

Figure 10. N₂ and CO₂ adsorption-desorption isotherms of CuNi SAs/UiO-66(Hf) and UiO-66(Hf). S_{BET} , pore volume and pore diameter of CuNi SAs/UiO-66(Hf) and UiO-66(Hf).

Figure 11. CO₂ adsorption properties of HKUST-1 irradiated with different doses.

Reference

- 1 Ma, X. *et al.* Modulating coordination environment of single-atom catalysts and their proximity to photosensitive units for boosting MOF photocatalysis. *J. Am. Chem. Soc.* **143**, 12220-12229 (2021).
- 2 Abdel-Mageed, A. M. *et al.* Highly active and stable single-atom Cu catalysts supported by a metal–organic framework. *J. Am. Chem. Soc.* **141**, 5201-5210, (2019).
- 3 Tan, K. *et al.* Defect termination in the UiO-66 family of metal-organic frameworks: the role of water and modulator. *J. Am. Chem. Soc.* **143**, 6328-6332 (2021).
- 4 Shearer, G. C. *et al.* Defect engineering: tuning the porosity and composition of the metal-organic framework UiO-66 via modulated synthesis. *Chem. Mater.* **28**, 3749-3761 (2016).
- 5 Wu, H.; Chua, Y. S.; Krungleviciute, V.; Tyagi, M.; Chen, P.; Yildirim, T.; Zhou, W., Unusual and highly tunable missing-linker defects in zirconium metal–organic framework UiO-66 and their important effects on gas adsorption. *J. Am. Chem. Soc.* **135**, 10525-10532 (2013).
- 6 Volkringer, C. *et al.* Stability of metal-organic frameworks under gamma irradiation. *Chem. Commun.* **52**, 12502-12505 (2016).

REVIEWERS' COMMENTS

Reviewer #1 (Remarks to the Author):

The authors well resolved my concerns. I suggest the acceptance of this manuscript at its current form.

Reviewer #2 (Remarks to the Author):

I have read the revised manuscript and I am satisfied that it is ready for publication. Very nice work.

Reviewer #3 (Remarks to the Author):

I thank the authors for their thorough revisions, which I believe have mainly answered the comments of all four reviewers satisfactorily. This manuscript is suitable for publication after the following remaining minor issues are addressed:

1. Fig. 1 caption: Change "Schema of radiation-catalyzed CO₂" to "Schema of radiation-catalyzed CO₂ conversion".
2. Fig. 5b and associated text: Regarding the assignment of the 360 nm band to CO₂•⁻, it might be good to cite a literature paper to backup this assignment.
3. Supplementary Fig. 17: The caption for this figure is very poor. It should indicate that the spectra were recorded following pulse radiolysis, and it should state the exact conditions, i.e., CO₂-saturated aqueous 0.1 M formate solution.
4. Supplementary Information, Table 1: In the title of this table, change "Cu and loadings" to "Cu and Ni loadings".

Reviewer #4 (Remarks to the Author):

The authors diligently addressed the majority of the concerns raised and the revised manuscript is now recommended for publication.

Response to Reviewers' Comments

Reviewer #1 (Remarks to the Author):

The authors well resolved my concerns. I suggest the acceptance of this manuscript at its current form.

Response: We thank the reviewer for the recommendation of publication.

Reviewer #2 (Remarks to the Author):

I have read the revised manuscript and I am satisfied that it is ready for publication. Very nice work.

Response: We thank the reviewer for the recommendation of publication.

Reviewer #3 (Remarks to the Author):

I thank the authors for their thorough revisions, which I believe have mainly answered the comments of all four reviewers satisfactorily. This manuscript is suitable for publication after the following remaining minor issues are addressed:

Response: We thank the reviewer for the recommendation of publication.

1. Fig. 1 caption: Change "Schema of radiation-catalyzed CO₂" to "Schema of radiation-catalyzed CO₂ conversion".

Response: We have changed the Fig. 1 caption.

2. Fig. 5b and associated text: Regarding the assignment of the 360 nm band to CO₂^{•-}, it might be good to cite a literature paper to backup this assignment.

Response: Thank for your correction. We have cited the literature to illustrate this problem.

3. Supplementary Fig. 17: The caption for this figure is very poor. It should indicate that the spectra were recorded following pulse radiolysis, and it should state the exact conditions, i.e., CO₂-saturated aqueous 0.1 M formate solution.

Response: Thank for your correction. We have changed the caption of this figure.

4. Supplementary Information, Table 1: In the title of this table, change "Cu and loadings" to "Cu and Ni loadings".

Response: We have revised this error.

Reviewer #4 (Remarks to the Author):

The authors diligently addressed the majority of the concerns raised and the revised manuscript is now recommended for publication.

Response: We thank the reviewer for the recommendation of publication.